# Learning to Assist Humans without Inferring Rewards

**Vivek Myers**[1]     **Evan Ellis**[1]

**Sergey Levine**[1]     **Benjamin Eysenbach**[2]     **Anca Dragan**[1]

[1]UC Berkeley     [2]Princeton University

## Abstract

Assistive agents should make humans' lives easier. Classically, such assistance is studied through the lens of inverse reinforcement learning, where an assistive agent (e.g., a chatbot, a robot) infers a human's intention and then selects actions to help the human reach that goal. This approach requires inferring intentions, which can be difficult in high-dimensional settings. We build upon prior work that studies assistance through the lens of empowerment: an assistive agent aims to maximize the influence of the human's actions such that they exert a greater control over the environmental outcomes and can solve tasks in fewer steps. We lift the major limitation of prior work in this area — scalability to high-dimensional settings — with contrastive successor representations. We formally prove that these representations estimate a similar notion of empowerment to that studied by prior work and provide a mechanism for optimizing it. Empirically, our proposed method outperforms prior methods on synthetic benchmarks, and scales to Overcooked, a cooperative game setting. Theoretically, our work connects ideas from information theory, neuroscience, and reinforcement learning, and charts a path for representations to play a critical role in solving assistive problems.[1]

## 1  Introduction

AI agents deployed in the real world should be helpful to humans. When we know the utility function of the humans an agent could interact with, we can directly train assistive agents through reinforcement learning with the known human objective as the agent's reward. In practice, agents rarely have direct access to a scalar reward corresponding to human preferences (if such a consistent model even exists) [1], and must infer them from human behavior [2, 3]. This inference can be challenging, as humans may act suboptimally with respect to their stated goals, not know their goals, or have changing preferences [4]. Optimizing a misspecified reward function can have poor consequences [5].

An alternative paradigm for assistance is to train agents that are *intrinsically* motivated to assist humans, rather than directly optimizing a model of their preferences. An analogy can be drawn to a parent raising a child. A good parent will empower the child to make impactful decisions and flourish, rather than proscribing an "optimal" outcome for the child. Likewise, AI agents might seek to *empower* the human agents they interact with, maximizing their capacity to change the environment [6]. In practice, concrete notions of empowerment can be difficult to optimize as an objective, requiring extensive modeling assumptions that don't scale well to the high-dimensional settings deep reinforcement learning agents are deployed in.

What is a good intrinsic objective for assisting humans that doesn't require these assumptions? We propose a notion of assistance based on maximizing the influence of the human's actions on the

---

[1]Code: `https://github.com/vivekmyers/empowerment_successor_representations`
Website: `https://empowering-humans.github.io`

38th Conference on Neural Information Processing Systems (NeurIPS 2024).

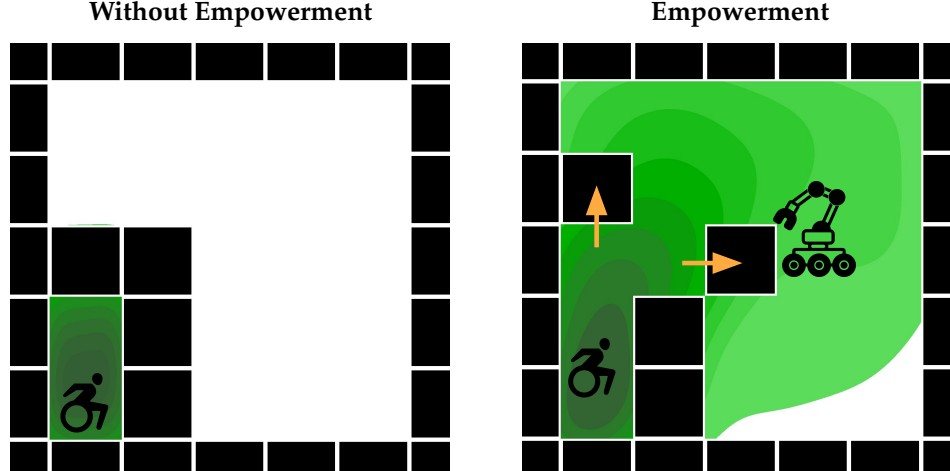

**Without Empowerment**  **Empowerment**

Figure 1: We propose an algorithm training assistive agents to empower human users - the assistant should take actions that enable human users to visit a wide range of future states, and the human's actions should exert a high degree of influence over the future outcomes. Our algorithm scales to high-dimensional settings, opening the door to building assistive agents that need not directly reason about human intentions.

environment. This approach only requires one structural assumption: the AI agent is interacting with an environment where there is a notion of actions taken by the human agent — a more general setting than the case where we model the human actions as the outcome of some optimization procedure, as in inverse RL [7, 8] or preference-based RL [9].

Prior work has studied various objectives for empowerment. For instance, Du et al. [6] approximate human empowerment as the variance in the final states of random rollouts. Despite excellent results in certain settings, this approach can be challenging to scale to higher dimensional settings, and does not necessarily enable human users to achieve the goals the want to achieve. By contrast, our approach exclusively empowers the human with respect to the distribution of (useful) behaviors induced by their current policy (which we term the *effective empowerment*), and can be implemented through a simple objective derived from contrastive successor features, which can then be optimized with scalable deep reinforcement learning (Fig. 1). We provide a theoretical framework connecting our objective to prior work on empowerment and goal inference, and empirically show that agents trained with this objective can assist humans in the Overcooked environment [10] as well as more complex versions of the obstacle gridworld assistance benchmark proposed by Du et al. [6].

Our core contribution is an objective for training agents that are intrinsically motivated to assist humans without requiring a model of the human's reward function. The approach, Empowerment via Successor Representations (ESR), maximizes the influence of the human's actions on the environment, and, unlike past approaches for assistance without reward inference, is based on a scalable model-free objective that can be derived from learned successor features that encode which states the human may want to reach given their current action. By maximizing effective empowerment, our objective empowers the human to reach the desired states, not all states, without assuming a human model. We analyze this objective in terms of empowerment and goal inference, drawing novel mathematical connections between time-series representations, decision-making, and assistance. We empirically show that agents trained with our objective can assist humans in two benchmarks proposed by past work: the Overcooked environment [10] and an obstacle-avoidance gridworld [6].

## 2 Related Work

Our approach broadly connects ideas from contrastive representation learning and intrinsic motivation to the problem of assisting humans.

**Assistive Agents.**    There are two lines of past work on assistive agents that are most relevant.

The first line of work focuses on the setting of an assistance game [2], where a robot (AI) agent tries to optimize a human reward of which it is initially unaware. Practically, inverse reinforcement learning (IRL) can be used in such a setting to infer the human's reward function and assist the human in achieving their goals [3]. The key challenge with this approach is that it requires modeling the human's reward function. This can be difficult in practice, especially if the human's behavior is not well-modeled by the reward architecture. Slightly misspecified reward functions can lead to catastrophic outcomes (i.e., directly harmful behavior in the assistance context) [11–13]. These concerns have prompted interest, under the umbrella term *AI Alignment*, in ensuring agents are safe and adhere to human values [14–17]. Our approach avoids some of these issues by focusing on empowerment, a more general objective that does not require modeling the human's reward function.

The second line of work focuses on empowerment-like objectives for assistance and shared autonomy. Empowerment generally refers to a measure of an agent's ability to influence the environment [18, 19]. In the context of assistance, Du et al. [6] show one such approximation of empowerment (AvE) can be approximated in simple environments through random rollouts to assist humans. Meanwhile, empowerment-like objectives have been used in shared autonomy settings to assist humans with teleoperation [20] and general assistive interfaces [21]. A key limitation of these approaches for general assistance is they only model empowerment over one time step. Our approach enables a more scalable notion of empowerment that can be computed over multiple time steps.

**Intrinsic Motivation.**   Intrinsic motivation broadly refers to agents that accomplish behaviors in the absence of an externally-specified reward or task [22]. Common applications of intrinsic motivation in single-agent reinforcement learning include exploration and skill discovery [23–25], empowerment [18, 19], and surprise minimization [19, 26, 27]. When applied to settings with humans, these objectives may lead to antisocial behavior [5]. Our approach applies intrinsic motivation to the setting of assisting humans, where the agent's goal is an empowerment objective — to maximize the human's ability to change the environment.

**Information-theoretic Decision Making.**   Information-theoretic approaches have seen broad applicability across unsupervised reinforcement learning [19, 23, 28]. These methods have been applied to goal-reaching [29], skill discovery [24, 30–33], and exploration [25, 34, 35]. In the context of assisting humans, information-theoretic methods have primarily been used to reason about the human's goals or rewards [36–38].

Our approach is made possible by advances in contrastive representation learning for efficient estimation of the mutual information of sequence data [39]. While these methods have been widely used for representation learning [40, 41] and reinforcement learning [42–45], to the best of our knowledge prior work has not used these contrastive techniques for learning assistive agents.

## 3   The Information Geometry of Empowerment

We will first state a general notion of an assistive setting, then show how an empowerment objective based on learned successor representations can be used to assist humans without making assumptions about the human following an underlying reward function. In Section 5, we provide empirical evidence supporting these claims.

### 3.1   Preliminaries

Formally, we adapt the notation of Hadfield-Menell et al. [2], and assume a "robot" ($\mathbf{R}$) and "human" ($\mathbf{H}$) policy are training together in an MDP $M = (\mathcal{S}, \mathcal{A}_{\mathbf{H}}, \mathcal{A}_{\mathbf{R}}, R, \mathrm{P}, \gamma)$. The states $s$ consist of the joint states of the robot and the human; we do not have separate observations for the human and robot. At any state $s \in \mathcal{S}$, the robot policy selects actions distributed according to $\pi_{\mathbf{R}}(a^{\mathbf{R}} \mid s)$ for $a^{\mathbf{R}} \in \mathcal{A}_{\mathbf{R}}$ and the human selects actions from $\pi_{\mathbf{H}}(a^{\mathbf{H}} \mid s)$ for $a^{\mathbf{H}} \in \mathcal{A}_{\mathbf{H}}$. The transition dynamics are defined by a distribution $\mathrm{P}(s' \mid s, a^{\mathbf{H}}, a^{\mathbf{R}})$ over the next state $s' \in \mathcal{S}$ given the current state $s \in \mathcal{S}$ and actions $a^{\mathbf{H}} \in \mathcal{A}_{\mathbf{H}}$ and $a^{\mathbf{R}} \in \mathcal{A}_{\mathbf{R}}$, as well as an initial state distribution $\mathrm{P}(s_0)$. For notational convenience, we will additionally define random variables $\mathfrak{s}_t$ to represent the state at time $t$, and $\mathfrak{a}_t^{\mathbf{R}} \sim \pi_{\mathbf{R}}(\bullet \mid \mathfrak{s}_t)$ and $\mathfrak{a}_t^{\mathbf{H}} \sim \pi_{\mathbf{H}}(\bullet \mid \mathfrak{s}_t)$ to represent the human and robot actions at time $t$, respectively.

**Effective Empowerment.**   Our work builds on a long line of prior methods that use information theoretic objectives for RL. Specifically, we adopt *empowerment* as an objective for training an assistive agent [6, 18, 46]. This section provides the mathematical foundations for empowerment,

as developed in prior work. We build on the prior work by (1) providing an information geometric interpretation of what empowerment does (Section 3.3) and (2) providing a scalable algorithm for estimating and optimizing a notion of empowerment, the *effective empowerment* (Section 4.1).

The idea behind empowerment is to think about the changes that an agent can effect on a world; an agent is more empowered if its actions effect a larger degree of change over future outcomes. Following prior work [18, 29, 46], we measure empowerment by looking at how much the actions taken *now* affect outcomes *in the future*. An agent with a high degree of empowerment exerts a high degree of control of the future states by simply changing its current actions. Like prior work, we measure this degree of control through the mutual information $I(\mathfrak{s}^+; \mathfrak{a}^{\mathbf{H}})$ between the current action $\mathfrak{a}^{\mathbf{H}}$ and the future states $\mathfrak{s}^+$. Note that these future states might occur many time steps into the future.

Empowerment depends on several factors: the environment dynamics, the choice of future actions, the current state, and other agents in the environment. Different problem settings involve maximizing empowerment using these different factors. In this work, we study the setting where a "human" agent and a "robot" agent collaborate in an environment, with the robot aiming to maximize the empowerment of the human. This problem setting was introduced in prior work [2, 6]. Compared with other mathematical frameworks for learning assistive agents [47], framing the problem in terms of empowerment means that the assistive agent need not infer the human's underlying intention, an inference problem that is typically challenging [48, 49].

To define our empowerment objective, we introduce the random variable $\mathfrak{s}^+$, corresponding to a state sampled $K \sim \mathrm{Geom}(1 - \gamma)$ steps into the future under the behavior policies $\pi_{\mathbf{H}}$ and $\pi_{\mathbf{R}}$. We will use $\rho(\mathfrak{s}^+ \mid s_t)$ to denote the density of this random variable; this density is sometimes referred to as the discounted state occupancy measure. We will use mutual information to measure how much the action $a_t$ at time $t$ changes this distribution:

$$I(\mathfrak{a}_t^{\mathbf{H}}; \mathfrak{s}^+ \mid s_t) \triangleq \mathbb{E}_{s_t, s_{t+k}, a_t^{\mathbf{H}}, a_t^{\mathbf{R}}} \left[ \log \frac{\mathrm{P}(\mathfrak{s}_{t+K} = s_{t+k} \mid \mathfrak{s}_t = s_t, \mathfrak{a}_t^{\mathbf{H}} = a_t)}{\mathrm{P}(\mathfrak{s}_{t+K} = s_{t+k} \mid \mathfrak{s}_t = s_t)} \right]. \tag{1}$$

Our overall objective is *effective empowerment*, $\mathcal{E}(\pi_{\mathbf{H}}, \pi_{\mathbf{R}})$: the mutual information between the human's actions and the future states $\mathfrak{s}^+$ while interacting with the robot:

$$\mathcal{E}(\pi_{\mathbf{H}}, \pi_{\mathbf{R}}) = \mathbb{E} \left[ \sum_{t=0}^{\infty} \gamma^t I(\mathfrak{a}_t^{\mathbf{H}}; \mathfrak{s}^+ \mid \mathfrak{s}_t) \right]. \tag{2}$$

We call this objective *effective empowerment* rather than just *empowerment* to highlight that the mutual information in Eq. (2) is computed under the behavior policies $\pi_{\mathbf{H}}$ and $\pi_{\mathbf{R}}$. This is in contrast to much of the prior work on empowerment, which focuses on the highest possible empowerment that could be achieved under some policy, rather than the actual realized empowerment [6, 18, 46]. Focusing on the effective empowerment simplifies the problem of estimating empowerment since we only have access to samples from the current behavior policies, not arbitrary more powerful policies. In later sections, we will see this simplification is both empirically effective and theoretically justified.

Note that this objective resembles an RL objective: we do not just want to maximize this objective greedily at each time step, but rather want the assistive agents to take actions now that help the human agent reach states where it will have high empowerment in the future.

## 3.2 Intuition and Geometry of Empowerment

Intuitively, the assistive agent should aim to maximize the number of future that can be realized by the human's actions. We will mathematically quantify this in terms of the discounted state occupancy measure, $\rho^{\pi}(s^+ \mid s)$. An agent has a large empowerment if the future states for one action are very different from the future actions after taking a different action; i.e., when $\rho(a_t = a_1; \mathfrak{s}^+ \mid s_t)$ is quite different from $\rho(a_t \mid s_2; \mathfrak{s}^+ \mid s_t)$ for actions $a_1 \neq a_2$. The mutual information (Eq. 1) quantifies this degree of control: $I(\mathfrak{a}_t; \mathfrak{s}^+ \mid s_t)$.

One way of understanding this mutual information is through *information geometry* [50, 51, 51, 52]. For a fixed current state $s_t$, assistant policy $\pi_{\mathbf{R}}$ and human policy $\pi_{\mathbf{H}}$, each potential action $a_t$ that the human takes induces a different distribution over future states: $\rho^{\pi_{\mathbf{R}}, \pi_{\mathbf{H}}}(\mathfrak{s}^+ \mid s_t, a_t)$. We can think about the set of these possible distributions: $\{\rho^{\pi_{\mathbf{R}}, \pi_{\mathbf{H}}}(\mathfrak{s}^+ \mid s_t, a_t) \mid a_t \in \mathcal{A}\}$. Figure 2 *(Left)* visualizes this distribution on a probability simplex for 6 choices of action $a_t$. If we look at any possible distribution over actions, then this set of possible future distributions becomes a polytope (see orange polygon in Fig. 2 *(Center)*).

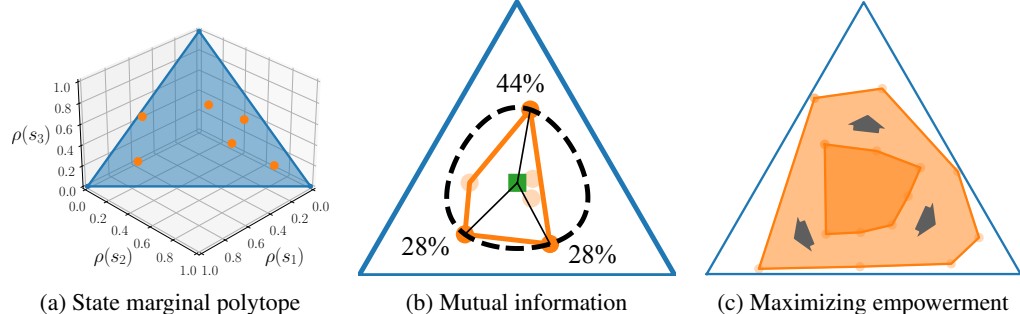

|(a) State marginal polytope | (b) Mutual information | (c) Maximizing empowerment |

Figure 2: **The Information Geometry of Empowerment**, illustrating the analysis in Section 3.3. *(Left)* For a given state $s_t$ and assistant policy $\pi_\mathbf{R}$, we plot the distribution over future states for 6 choices of the human policy $\pi_\mathbf{H}$. In a 3-state MDP, we can represent each policy as a vector lying on the 2-dimensional probability simplex. We refer to the set of all possible state distributions as the *state marginal polytope*. *(Center)* Mutual information corresponds to the distance between the center of the polytope and the vertices that are maximally far away. *(Right)* Empowerment corresponds to maximizing the size of this polytope. For example, when an assistive agent moves an obstacle out of a human user's way, the human user can spend more time at desired state.

Intuitively, the mutual information $I(\mathfrak{a}_t; \mathfrak{s}^+ \mid s_t)$ in our empowerment objective corresponds to the *size* or *volume* of this state marginal polytope. This intuition can be formalized by using results from information geometry [53–55]. The human policy $\pi_\mathbf{H}(a_t \mid s_t)$ places probability mass on the different points in Figure 2 (Center). Maximizing the mutual information corresponds to "picking out" the state distributions that are maximally spread apart (see probabilities in Fig. 2 *(Center)*). To formalize this, define

$$\rho(\mathfrak{s}^+ \mid s_t) \triangleq \mathbb{E}_{\pi(a_t \mid s_t)}[\rho(\mathfrak{s}^+ \mid s_t, a_t)] \tag{3}$$

as the *average* state distribution from taking the human's actions (see green square in Fig. 2 *(Center)*).

**Remark 3.1.** *Mutual information corresponds to the distance between the average state distribution (Eq. 3) and the furthest achievable state distributions:*

$$I(\mathfrak{a}_t; \mathfrak{s}^+ \mid s_t) = \max_{a_t} D_{KL}\big(\rho(a_t; \mathfrak{s}^+ \mid s_t) \,\|\, \rho(\mathfrak{s}^+ \mid s_t)\big) \triangleq d_{max}. \tag{4}$$

This distance is visualized as the black lines in Fig. 2. When we talk about the "size" of the state marginal polytope, we are specifically referring to the length of these black lines (as measured with a KL divergence).

This sort of mutual information is a way for measuring the degree of control that an agent exerts on an environment. This measure is well defined for any agent/policy; that agent need not be maximizing mutual information, and could instead be maximizing some arbitrary reward function. This point is important in our setting: this means that the assistive agent can estimate and maximize the empowerment of the human user *without having to infer what reward function the human is trying to maximize.*

Finally, we come back to our empowerment objective, which is a discounted sum of the mutual information terms that we have been analyzing above. This empowerment objective says that the human is more empowered when this set has a larger size — i.e., the human can visit a wider range of future state (distributions). The empowerment objective says that the assistive agent should act to try to maximize the size of this polytope. Importantly, this maximization problem is done sequentially: the assistive agent wants the size of this polytope to be large both at the current state and at future states; the human's actions should exert a high degree of influence over the future outcomes both now and in the future. Thus, our overall objective looks at a sum of these mutual informations.

Not only does this analysis provides a geometric picture for what empowerment is doing, it also lays the groundwork for formally relating empowerment to reward.

## 3.3 Relating Empowerment to Reward

In this section we take aim at the question: when humans are well-modeled as optimizing a reward function, when does maximizing effective empowerment help humans maximize their rewards? Answering this question is important because for empowerment to be a safe and effective assistive objective, it should enable the human to better achieve their goals. We show that under certain assumptions, empowerment yields a provable lower bound on the average-case reward achieved by the human for suffiently long-horizon empowerment (i.e., $\gamma \to 1$).

For constructing the formal bound, we suppose the human is Boltzmann-rational [56, 57] with respect to some reward function $R \sim \mathcal{R}$ and rationality coefficient $\beta$. The distribution $\mathcal{R}$ could be interpreted as a prior over the human's objective, a set of skills the human may try and carry out, or a population of humans with different objectives that the agent could be interacting with. Our quantity of interest, the average-case reward achieved by the human with our empowerment objective, is given by

$$\mathcal{J}_{\pi_{\mathbf{R}}}^{\gamma}(\pi_{\mathbf{H}}) = \mathbb{E}_{\substack{R \sim \mathcal{R} \\ s_0 \sim p_0}} \left[ V_{R,\gamma}^{\pi_{\mathbf{H}}, \pi_{\mathbf{R}}}(s_0) \right] \tag{5}$$

where $V_{R,\gamma}^{\pi_{\mathbf{H}}, \pi_{\mathbf{R}}}(s_0)$ is the value function of the human policy $\pi_{\mathbf{H}}$ under the reward function $R$ when interacting with $\pi_{\mathbf{R}}$. Recalling Eq. (2), we will express the overall effective empowerment objective we are trying to relate to Eq. (5) as

$$\mathcal{E}_{\gamma}(\pi_{\mathbf{H}}, \pi_{\mathbf{R}}) = \mathbb{E}\left[ \sum_{t=0}^{\infty} \gamma^t I(\mathfrak{s}^+; \mathfrak{a}_t^{\mathbf{H}} \mid \widetilde{\mathfrak{s}}_t) \right]. \tag{6}$$

This notation is formalized in Appendix B.

The two key assumptions used in our analysis are Assumption 3.1, which states that the human will optimize for behaviors that uniformly cover the state space, and Assumption 3.2, which simply states that with infinite time, the human will be able to reach any state in the state space.

**Assumption 3.1** (Skill Coverage). *The rewards $R \sim \mathcal{R}$ are uniformly distributed over the scaled $|\mathcal{S}|$-simplex $\Delta^{|\mathcal{S}|}$ such that:*

$$\left(R + \tfrac{1}{|\mathcal{S}|}\right)\left(\tfrac{1}{1-\gamma}\right) \sim \mathrm{Unif}\left(\Delta^{|\mathcal{S}|}\right) = \mathrm{Dirichlet}(\underbrace{1, 1, \ldots, 1}_{|\mathcal{S}| \text{ times}}). \tag{7}$$

**Assumption 3.2** (Ergodicity). *For some $\pi_{\mathbf{H}}, \pi_{\mathbf{R}}$, we have*

$$\mathrm{P}^{\pi_{\mathbf{H}}, \pi_{\mathbf{R}}}(\mathfrak{s}^+ = s \mid s_0) > 0 \quad \text{for all } s \in \mathcal{S}, \gamma \in (0, 1). \tag{8}$$

Our main theoretical result is Theorem 3.1, which shows that under these assumptions, maximizing effective empowerment yields a lower bound on the (squared) average-case reward achieved by the human for sufficiently large $\gamma$. In other words, for a sufficiently long empowerment horizon, the empowerment objective Eq. (2) is a meaningful proxy for reward maximization.

**Theorem 3.1.** *Under Assumption 3.1 and Assumption 3.2, for sufficiently large $\gamma$ and any $\beta > 0$,*

$$\mathcal{E}_{\gamma}(\pi_{\mathbf{H}}, \pi_{\mathbf{R}})^{1/2} \leq (\beta/e) \, \mathcal{J}_{\pi_{\mathbf{R}}}^{\gamma}(\pi_{\mathbf{H}}). \tag{9}$$

The proof is in Appendix B.1 To the best of our knowledge, this result provides the first formal link between empowerment maximization and reward maximization. This motivates us to develop a scalable algorithm for empowerment maximization, which we introduce in the following section.

## 4 Estimating and Maximizing Empowerment with Contrastive Representations

Directly computing Eq. (2) would require access to the human policy, which we don't have. Therefore, we want a tractable estimation that still performs well in large environments which are more difficult to model due to the exponentially increasing set of possible future states. To better-estimate empowerment, we learn contrastive representations that encode information about which future states are likely to be reached from the current state. These contrastive representations learn to model mutual information between the current state, action, and future state, which we then use to compute the empowerment objective.

## 4.1 Estimating Effective Empowerment

To estimate this effective empowerment objective, we need a way of learning the probability ratio inside the expectation. Prior methods such as Du et al. [6] and Salge et al. [18] rollout possible future states and compute a measure of their variance as a proxy for empowerment, however this doesn't scale when the environment becomes complex. Other methods learn a dynamics model, which also doesn't scale when dynamics become challenging to model [31]. Modeling these probabilities directly is challenging in settings with high-dimensional states, so we opt for an indirect approach. Specifically, we will learn representations that encode two probability ratios. Then, we will be able to compute the desired probability ratio by combining these other probability ratios.

Our method learns three representations:

1. $\phi(s, a^{\mathbf{R}}, a^{\mathbf{H}})$ — This representation can be understood as a sort of latent-space model, predicting the future representation given the current state $s$ and the human's current action $a^{\mathbf{H}}$ as well as the robot's current action $a^{\mathbf{R}}$.

2. $\phi'(s, a^{\mathbf{R}})$ — This representation can be understood as an uncontrolled model, predicting the representation of a future state without reference to the current human action $a^{\mathbf{H}}$. This representation is analogous to a value function.

3. $\psi(s^+)$ — This is a representation of a future state.

We will learn these three representations with two contrastive losses, one that aligns $\phi(s, a^R, a^H) \leftrightarrow \psi(s^+)$ and one that aligns $\phi'(s, a^{\mathbf{R}}) \leftrightarrow \psi(s^+)$

$$\max_{\phi, \phi', \psi} \mathbb{E}_{\{(s_i, a_i, s'_i) \sim P(\mathfrak{s}_t, \mathfrak{a}_t^{\mathbf{H}}, \mathfrak{s}_{t+k})\}_{i=1}^N} \left[ \mathcal{L}_{\mathsf{c}}(\{\phi(s_i, a_i)\}, \{\psi(s'_j)\}) + \mathcal{L}_{\mathsf{c}}(\{\phi'(s_i)\}, \{\psi(s'_j)\}) \right],$$

where the contrastive loss $\mathcal{L}_{\mathsf{c}}$ is the symmetrized infoNCE objective [39]:

$$\mathcal{L}_{\mathsf{c}}(\{x_i\}, \{y_j\}) \triangleq \sum_{i=1}^N \left[ \log\left( \frac{e^{x_i^T y_i}}{\sum_{j=1}^N e^{x_i^T y_j}} \right) + \log\left( \frac{e^{x_i^T y_i}}{\sum_{j=1}^N e^{x_j^T y_i}} \right) \right]. \tag{10}$$

We have colored the index $j$ for clarity. At convergence, these representations encode two probability ratios [28], which we will ultimately be able to use to estimate empowerment (Eq. 2):

$$\phi(s, a^{\mathbf{R}}, a^{\mathbf{H}})^T \psi(g) = \log\left[ \frac{P(\mathfrak{s}_{t+K} = g \mid \mathfrak{s}_t = s, \mathfrak{a}_t^{\mathbf{H}} = a^{\mathbf{H}}, \mathfrak{a}_t^{\mathbf{R}} = a^{\mathbf{R}})}{C_1 P(\mathfrak{s}_{t+K} = g)} \right] \tag{11}$$

$$\phi'(s, a^{\mathbf{R}})^T \psi(g) = \log\left[ \frac{P(\mathfrak{s}_{t+K} = s_{t+k} \mid \mathfrak{s}_t = s_t, \mathfrak{a}_t^{\mathbf{R}} = a^{\mathbf{R}})}{C_2 P(\mathfrak{s}_{t+K} = g)} \right]. \tag{12}$$

Note that our definition of empowerment (Eq. 2) is defined in terms of similar probability ratios. The constants $C_1$ and $C_2$ will mean that our estimate of empowerment may be off by an additive constant, but that constant will not affect the solution to the empowerment maximization problem.

## 4.2 Estimating Empowerment with the Learned Representations

To estimate empowerment, we will look at the difference between these two inner products:

$$\phi(s_{t+K}, a^{\mathbf{R}}, a^{\mathbf{H}})^T \psi(g) - \phi(s_{t+K}, a^{\mathbf{R}})^T \psi(g)$$
$$= \log P(s_{t+K} \mid s, a^{\mathbf{H}}) - \log C_1 - \underline{\log P(s_{t+K})} - \log P(s_{t+K} \mid s) + \log C_2 + \underline{\log P(s_{t+K})}$$
$$= \log \frac{P(s_{t+K} \mid s, a^{\mathbf{H}})}{P(s_{t+K} \mid s)} + \log \frac{C_2}{C_1}.$$

Note that the expected value of the first term is the *conditional* mutual information $I(s_{t+K}; a^{\mathbf{H}} \mid s)$. Our empowerment objective corresponds to averaging this mutual information across all the visited states. In other words, our objective corresponds to an RL problem, where empowerment corresponds to the expected discounted sum of these log ratios:

$$\mathcal{E}(\pi_{\mathbf{H}}, \pi_{\mathbf{R}}) = \mathbb{E}_{\pi_{\mathbf{H}}, \pi_{\mathbf{R}}} \left[ \sum_{t=0}^{\infty} \gamma^t I(s^+; a_t^{\mathbf{H}} \mid s_t) \right]$$

$$\approx \mathbb{E}_{\pi_{\mathbf{H}}, \pi_{\mathbf{R}}} \left[ \sum_{t=0}^{\infty} \gamma^t (\phi(s_t, a^{\mathbf{R}}, a^{\mathbf{H}}) - \phi(s_t, a^{\mathbf{R}}))^T \psi(g) - \log \frac{C_2}{C_1} \right].$$

The approximation above comes from function approximation in learning the Bayes optimal representations. Again, note that the constants $C_1$ and $C_2$ do not change the optimization problem. Thus, to maximize empowerment we will apply RL to the assistive agent $\pi_\mathbf{R}(a \mid s)$ using a reward function

$$r(s, a^\mathbf{R}) = (\phi(s_t, a^\mathbf{R}, a^\mathbf{H}) - \phi(s_t, a^\mathbf{R}))^T \psi(g). \tag{13}$$

### 4.3 Algorithm Summary

---

**Algorithm 1:** Empowerment via Successor Representations (ESR)

---

**Input**: Human policy $\pi_\mathbf{H}(a \mid s)$
Randomly initialize assistive agent policy $\pi_\mathbf{R}(a \mid s)$, and representations $\phi(s, a^\mathbf{R}, a^\mathbf{H})$, $\psi(s, a^T)$, and $\psi(g)$.
Initialize replay buffer $\mathcal{B}$.
**while** not converged **do**
    Collect a trajectory of experience with human policy and assistive agent policy, store in replay buffer $\mathcal{B}$.
    Update representations $\phi(s, a^\mathbf{R}, a^\mathbf{H})$, $\psi(s, a^T)$, and $\psi(g)$ with the contrastive losses in Eq. (10).
    Update $\pi_\mathbf{R}(a \mid s)$ with RL using reward function $r(s, a^\mathbf{R}, a^\mathbf{H}) = (\phi(s, a^\mathbf{R}, a^\mathbf{H}) - \phi'(s, a^\mathbf{R}))^T \psi(g)$.
**Return**: Assistive policy $\pi_\mathbf{R}(a \mid s)$.

---

We propose an actor-critic method for learning the assistive agent. Our method will alternate between updating these contrastive representations and using them to estimate a reward function (Eq. (13)) that is optimized via RL. We summarize the algorithm in Algorithm 1. In practice, we use SAC [58] as our RL algorithm. In our experiments, we will also study the setting where the human user updates their policy alongside the assistive agent.

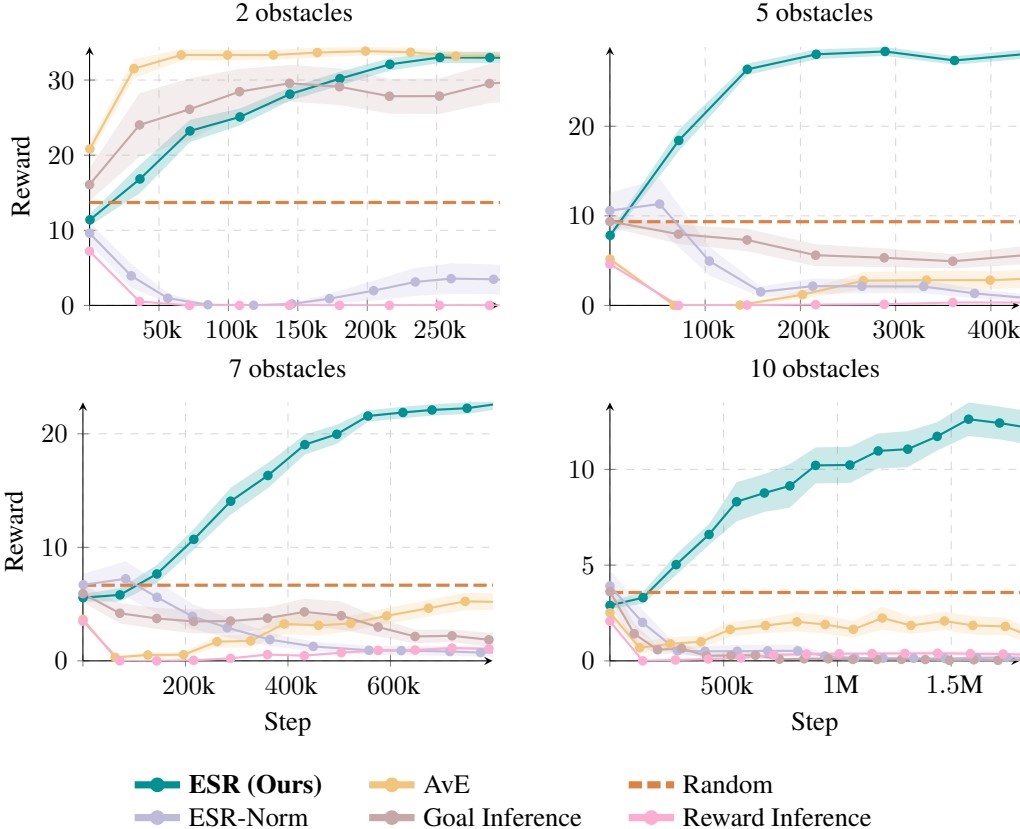

Figure 3: We apply our method to the benchmark proposed in prior work [6], visualized in Fig. 4a. The four subplots show variant tasks of increasing complexity (more blocks), ($\pm 1$ SE). We compare against AvE [6], the Goal Inference baseline from [6] which assumes access to a world model, and Reward Inference [59] where we recover the reward from a learned q-value. These prior approaches fail on all except the easiest task, highlighting the importance of scalability.

# 5 Experiments

We seek to answer two questions with our experiments. *First*, does our approach enable assistance in standard cooperation benchmarks? *Second*, does our approach scale to harder benchmarks where prior methods fail?

Our experiments will use two benchmarks designed by prior work to study assistance: the obstacle gridworld [6] and Overcooked [10]. Our main **baseline** is AvE [6], a prior empowerment-based method. Our conjecture is that both methods will perform well on the lower-dimensional grid-world task, and that our method will scale more gracefully to the higher dimensional Overcooked environment. We will also compare against a naïve baseline where the assistive agent acts randomly.

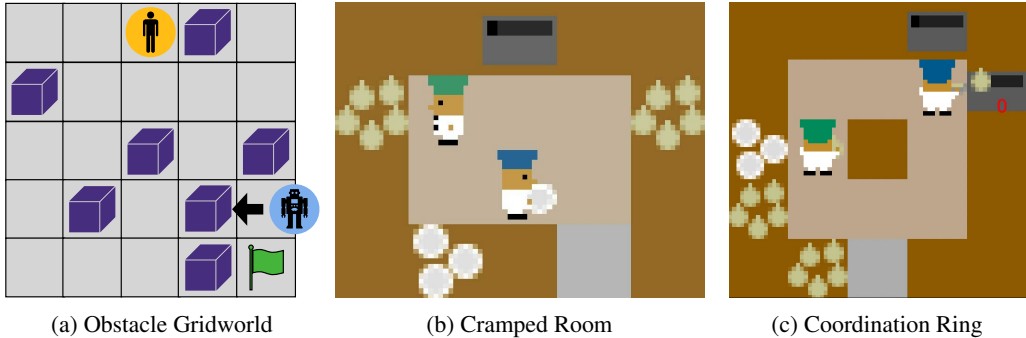

| (a) Obstacle Gridworld | (b) Cramped Room | (c) Coordination Ring |

Figure 4: *(a)* The modified environment from Du et al. [6] scaled to $N = 7$ blocks, and *(b, c)* the two layouts of the Overcooked environment [10].

## 5.1 Do contrastive successor representations effectively estimate empowerment?

We test our approach in the assistance benchmark suggested in Du et al. [6]. The human (orange) is tasked with reaching a goal state (green) while avoiding the obstacles (purple). The AI assistant can move blocks one step at a time in any direction [6]. While the original benchmark used $N = 2$ obstacles, we will additionally evaluate on harder versions of this task with $N = 5, 7, 10$ obstacles. We show results in Fig. 3. On the easiest task, both our method and AvE achieve similar asymptotic reward, though our method learns more slowly than AvE. However, on the tasks with moderate and high degrees of complexity, our approach (ESR) achieves significantly higher rewards than AvE, which performs worse than a random controller. These experiments support our claim that contrastive successor representations provide an effective means for estimating empowerment, and hint that ESR might be well suited for solving higher dimensional tasks.

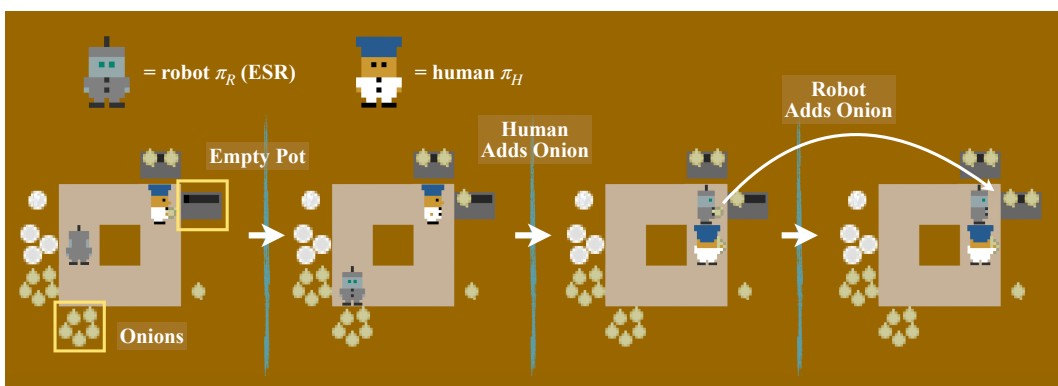

Figure 5: In Coordination Ring, our ESR agent learns to wait for the human to add an onion to the pot, and then adds one itself. There is another pot at the top which is nearly full, but the empowerment agent takes actions to maximize the impact of the human's actions, and so follows the lead of the human by filling the empty pot.

## 5.2 Does our approach scale to tasks with image-based observations?

Our second set of experiments look at scaling ESR to the image-based Overcooked environment. Since contrastive learning is often applied to image domains, we conjectured that ESR would scale gracefully to this setting. We will evaluate our approach in assisting a human policy trained with behavioral cloning taken from Laidlaw and Dragan [60]. The human prepares dishes by picking up ingredients and cooking them on a stove, while the AI assistant moves ingredients and dishes around the kitchen. We focus on two environments within this setting: a cramped room where the human must pass ingredients and dishes through a narrow corridor, and a coordination ring where the human must pass ingredients and dishes around a ring-shaped kitchen (Figs. 4b and 4c). As before, we compare with AvE as well as a naïve random controller. We report results in Table 1. On both tasks, we observe that our approach achieves higher rewards than AvE baseline, which performs no better than a random controller. In Fig. 5, we show an example of one of the collaborative behaviors learned by ESR. Taken together with the results in the previous setting, these results highlight the scalability of ESR to higher dimensional problems.

Table 1: Overcooked Results

| Layout | ESR (Ours) | Reward Inference | AvE | Random |
|---|---|---|---|---|
| Asymmetric Advantages | $72.00 \pm 5.37$ | $60.33 \pm 0.26$ | $36.71 \pm 1.71$ | $59.36$ |
| Coordination Ring | $8.40 \pm 0.69$ | $5.96 \pm 0.20$ | $5.69 \pm 0.93$ | $6.02$ |
| Cramped Room | $91.33 \pm 4.08$ | $39.24 \pm 0.35$ | $5.13 \pm 1.31$ | $69.26$ |

## 6 Discussion

One of the most important problems in AI today is equipping AI agents with the capacity to assist humans achieve their goals. While much of the prior work in this area requires inferring the human's intention, our work builds on prior work in studying how an assistive agent can *empower* a human user without inferring their intention. Relative to prior methods, we demonstrate how empowerment can be readily estimated using contrastive learning, paving the way for deploying these techniques on high-dimensional problems.

**Limitations.** One of the main limitations of our approach is the assumption that the assistive agent has access to the human's actions, which could be challenging to observe in practice. Automatically inferring the human's actions remains an important problem for future work. A second limitation is that the method is currently an on-policy method, in the sense that the assistive agent has to learn by trial and error. A third limitation is that the ESR formulation assumes that both agents share the same state space. In many cases the empowerment objective will still lead to desirable behavior, however, care must be taken in cases where the agent can restrict the information in its own observations, which could lead to reward hacking. Finally, our experiments do not test our method against real humans, whose policies may differ from the simulated policies. In the future, we plan to investigate techniques from off-policy evaluation and cooperative game theory to enable faster learning of assistive agents with fewer trials. We also plan to test the ESR objective in environments with partial observability over the human's state.

**Safety risks.** Perhaps the main risk involved with maximizing empowerment is that it may be at odds with a human's agents goal, especially in contexts where the pursuit of that goal limits the human's capacity to pursue other goals. For example, a family choosing to have a kid has many fewer options over where they can travel for vacation, yet we do not want assistive agents to stymie families from having children.

One key consideration is *whom* should be empowered. The present paper assumes there is a single human agent. Equivalently, this can be seen as maximizing the empowerment of all exogenous agents. However, it is easy to adapt the proposed method to maximize the empowerment of a single target individual. Given historical inequities in the distribution of power, practitioners must take care when considering whose empowerment to maximize. Similarly, while we focused on *maximizing* empowerment, it is trivial to change the sign so that an "assistive" agent minimizes empowerment. One could imagine using such a tool in policies to handicap one's political opponents.

**Acknowledgments.** We would like to thank Micah Carroll and Cameron Allen for their helpful feedback, as well as Niklas Lauffer for suggesting JaxMARL. We especially thank the fantastic NeurIPS reviewers for their constructive comments and suggestions. This research was partly supported by ARL DCIST CRA W911NF-17-2-0181 and ONR N00014-22-1-2773, as well as NSF HCC 2310757, the Jump Cocosys Center, Princeton Research Computing, and the DoD through the NDSEG Fellowship Program.

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

# A  Experimental Details

We ran all our experiments on NVIDIA RTX A6000 GPUs with 48GB of memory within an internal cluster. Each evaluation seed took around 5-10 hours to complete. Our losses (Eqs. 10 and 13) were computed and optimized in JAX with Adam [61]. We used a hardware-accelerated version of the Overcooked environment from the JaxMARL package [62]. The experimental results described in Section 5 were obtained by averaging over 5 seeds for the Overcooked coordination ring layout, 15 for the cramped room layout, and 20 for the obstacle gridworld environment. Specific hyperparameter values can be found in our code, which is available at `https://github.com/vivekmyers/empowerment_successor_representations`.

## A.1  Network Architecture

In the obstacle grid environment, we used a network with 2 convolutional and 2 fully connected layers and SiLU activations. In Overcooked, we adapted the policy architecture from past work [4], using 3 convolutional layers followed by 4 MLP layers with Leaky ReLU activations [63]. We concatenate in $a^{\mathbf{R}}$ and $a^{\mathbf{H}}$ to the state as one-hot encoded channels, i.e. if the action is 5, 6 additional channels will be concatenated to the state with all set to 0s except the 5th channel which is set to 1s.

# B  Theoretical Analysis of Empowerment

To connect our effective empowerment objective to reward, we will extend the notation in Section 3.1 to include a distribution over possible tasks the human might be trying to solve, $\mathcal{R}$, such that each $R \sim \mathcal{R}$ defines a distinct reward function $R : \mathcal{S} \to \mathbb{R}$. We assume $\pi_{\mathbf{R}}$ tries to maximize the $\gamma$-discounted effective empowerment of the human, defined as

$$\mathcal{E}_\gamma(\pi_{\mathbf{H}}, \pi_{\mathbf{R}}) = \mathbb{E}\left[\sum\nolimits_{t=0}^{\infty} \gamma^t I(\mathfrak{s}_+^\gamma; \mathfrak{a}_t^{\mathbf{H}} \mid \widetilde{\mathfrak{s}}_t)\right] \tag{Eq. 6}$$

for

$$\mathfrak{s}_+^\gamma \triangleq \big\{\mathfrak{s}_k \quad \text{for} \quad k \sim \text{Geom}(1-\gamma)\big\}. \tag{14}$$

We additionally define $\bar{\mathfrak{s}}_t$ to be the full history of states up to time $t$ and $\bar{\mathfrak{a}}_t^{\mathbf{H}}$ to be the full history of human actions up to time $t$,

$$\bar{\mathfrak{s}}_t = \{\mathfrak{s}_i\}_{i=0}^t,$$
$$\bar{\mathfrak{a}}_t^{\mathbf{H}} = \{\mathfrak{a}_i^{\mathbf{H}}\}_{i=0}^t. \tag{15}$$

Then, $\widetilde{\mathfrak{s}}_t$ is the full history of states and past human actions up to time $t$,

$$\widetilde{\mathfrak{s}}_t = \bar{\mathfrak{s}}_t \cup \bar{\mathfrak{a}}_{t-1}^{\mathbf{H}}. \tag{16}$$

Note that the definition of empowerment in Eq. (6) differs slightly from the original construction Eq. (2) — we condition on the full history of human actions, not just the most recent one. This distinction becomes irrelevant in practice if our MDP maintains history in the state, in which case we can equivalently use $\mathfrak{s}_t$ in place of $\widetilde{\mathfrak{s}}_t$.

Meanwhile, for any fixed $\pi_{\mathbf{R}}$ and $\beta > 0$, the human is Boltzmann-rational with respect to the robot's policy:

$$\pi_{\mathbf{H}}(a_t^{\mathbf{H}} \mid \widetilde{\mathfrak{s}}_t) \propto \exp\big(\beta Q_{R,\gamma}^{\pi_{\mathbf{H}}, \pi_{\mathbf{R}}}(s_t, a_t^{\mathbf{H}})\big) \tag{17}$$

$$\text{where} \quad Q_{R,\gamma}^{\pi_{\mathbf{H}}, \pi_{\mathbf{R}}}(s_t, a_t^{\mathbf{H}}) = \mathbb{E}\left[\sum\nolimits_{k=0}^{\infty} \gamma^k R(s_{t+k}) \; \Big| \; s_t, a_t^{\mathbf{H}}\right]. \tag{18}$$

Equivalently, we can define the human's (soft) Q-function and value as

$$Q_{R,\gamma}^{\pi_{\mathbf{H}}, \pi_{\mathbf{R}}}(s_t, a_t^{\mathbf{H}}) = R(s_t) + \gamma \mathbb{E}\left[V_{R,\gamma}^{\pi_{\mathbf{H}}, \pi_{\mathbf{R}}}(s_{t+1}) \; \Big| \; s_t, a_t^{\mathbf{H}}\right]$$

$$\text{for} \quad V_{R,\gamma}^{\pi_{\mathbf{H}}, \pi_{\mathbf{R}}}(s_t) = \mathbb{E}\left[R(s_t) + \gamma R(s_{t+1}) + \gamma^2 R(s_{t+2}) + \ldots \; \Big| \; s_t, a_t^{\mathbf{H}}\right]. \tag{19}$$

The overall human objective is to maximize the expected soft value:

$$\mathcal{J}_{\pi_{\mathbf{R}}}^\gamma(\pi_{\mathbf{H}}) = \mathbb{E}_{\substack{R \sim \mathcal{R} \\ s_0 \sim p_0}}\left[V_{R,\gamma}^{\pi_{\mathbf{H}}, \pi_{\mathbf{R}}}(s_0)\right]. \tag{Eq. 5}$$

Note that this definition of $\pi_{\mathbf{H}}$ depends on $R$ and $\pi_{\mathbf{R}}$ and is bounded $0 \le \mathcal{J}_{\pi_{\mathbf{R}}}^{\gamma}(\pi_{\mathbf{H}}) \le 1$. As in the CIRL setting [2], we assume robot is unable to access the true human reward $R : \mathcal{S} \to \mathbb{R}$. One way to think of the robot's task is as finding a Nash equilibrium between the objectives Eq. (6) and the human best response in Eq. (17).

For convenience, we will also define a multistep version of $Q_{R,\gamma}^{\pi_{\mathbf{H}},\pi_{\mathbf{R}}}$,

$$Q_{R,\gamma}^{\pi_{\mathbf{H}},\pi_{\mathbf{R}}}(s_t, a_t^{\mathbf{H}}, \dots, a_{t+K}^{\mathbf{H}}) = \mathbb{E}\Big[\sum_{k=0}^{\infty} \gamma^k R(\mathfrak{s}_{t+k}) \ \Big|\ s_t, a_t^{\mathbf{H}}, \dots, a_{t+K}^{\mathbf{H}}\Big]. \tag{20}$$

## B.1 Connecting Empowerment to Reward

Our approach will be to first relate the empowerment (influence of $\mathfrak{a}_t^{\mathbf{H}}$ on $\mathfrak{s}_+^{\gamma}$) to the mutual information between $\mathfrak{a}_t^{\mathbf{H}}$ and the reward $R$.

Then, we will connect this quantity to a notion of "advantage" for the human (Eq. 27), which in turn can be related to the expected reward under the human's policy. In its simplest form, this argument will require an assumption over the reward distribution:

**Assumption 3.1** (Skill Coverage). *The rewards $R \sim \mathcal{R}$ are uniformly distributed over the scaled $|\mathcal{S}|$-simplex $\Delta^{|\mathcal{S}|}$ such that:*

$$\big(R + \tfrac{1}{|\mathcal{S}|}\big)\big(\tfrac{1}{1-\gamma}\big) \sim \mathrm{Unif}\big(\Delta^{|\mathcal{S}|}\big) = \mathrm{Dirichlet}(\underbrace{1, 1, \dots, 1}_{|\mathcal{S}| \ times}). \tag{7}$$

In other words, Assumption 3.1 says our prior over the human's reward function is uniform with zero mean. This is not the only prior for which this argument works, but for general $\mathcal{R}$ we will need a correction term to incentivize states that are more likely across the distribution of $\mathcal{R}$. Another way to view Assumption 3.1 is that the human is trying to execute diverse "skills" $z \sim \mathrm{Unif}(\Delta^{|\mathcal{S}|})$.

We also assume ergodicity (Assumption 3.2). In the special case of an MDP that resets to some distribution with full support over $\mathcal{S}$, this assumption is automatically satisfied.

**Assumption 3.2** (Ergodicity). *For some $\pi_{\mathbf{H}}, \pi_{\mathbf{R}}$, we have*

$$\mathrm{P}^{\pi_{\mathbf{H}},\pi_{\mathbf{R}}}(\mathfrak{s}_+^{\gamma} = s \mid s_0) > 0 \quad \text{for all } s \in \mathcal{S}, \gamma \in (0,1). \tag{8}$$

Our main result connects empowerment directly to a (lower bound on) the human's expected reward.

**Theorem 3.1.** *Under Assumption 3.1 and Assumption 3.2, for sufficiently large $\gamma$ and any $\beta > 0$,*

$$\mathcal{E}_{\gamma}(\pi_{\mathbf{H}}, \pi_{\mathbf{R}})^{1/2} \le (\beta/e)\,\mathcal{J}_{\pi_{\mathbf{R}}}^{\gamma}(\pi_{\mathbf{H}}). \tag{9}$$

Theorem 3.1 says that for a long enough horizon (i.e., $\gamma$ close to 1), the robot's empowerment objective will lower bound the (squared, MaxEnt) human objective.

We make use of the following lemmas in the proof.

**Lemma B.1.** *For $t \sim \mathrm{Geom}(1 - \gamma)$ and any $K \ge 0$,*

$$\liminf_{\gamma \to 1} I(\mathfrak{s}_+^{\gamma}; \mathfrak{a}_t^{\mathbf{H}}, \dots, \mathfrak{a}_{t+K}^{\mathbf{H}} \mid \widetilde{\mathfrak{s}}_t) \le I(R; \mathfrak{a}_t^{\mathbf{H}}, \dots, \mathfrak{a}_{t+K}^{\mathbf{H}} \mid \widetilde{\mathfrak{s}}_t). \tag{21}$$

*Proof.* For sufficiently large $\gamma$, $\mathfrak{s}_+^{\gamma}$ will approach the stationary distribution of $\mathrm{P}^{\pi_{\mathbf{H}},\pi_{\mathbf{R}}}$ for a fixed $\pi_{\mathbf{H}}, \pi_{\mathbf{R}}$, irrespective of $\widetilde{\mathfrak{s}}_t$ and $\mathfrak{a}_t^{\mathbf{H}}, \dots, \mathfrak{a}_{t+K}^{\mathbf{H}}$ from Assumption 3.2. So,

$$\liminf_{\gamma \to 1} I(\mathfrak{s}_+^{\gamma}; \mathfrak{a}_t^{\mathbf{H}}, \dots, \mathfrak{a}_{t+K}^{\mathbf{H}} \mid \widetilde{\mathfrak{s}}_t) \le I\Big(\lim_{\gamma \to \infty} \mathfrak{s}_+^{\gamma}\, ; \mathfrak{a}_t^{\mathbf{H}}, \dots, \mathfrak{a}_{t+K}^{\mathbf{H}} \ \Big|\ \widetilde{\mathfrak{s}}_t\Big) \tag{22}$$

Since each $R, \pi_{\mathbf{R}}, \gamma$ defines a human policy $\pi_{\mathbf{H}}$ via Eq. (17), we can express the dependencies as the following Markov chain:

$$\hat{\mathfrak{a}}_t \longrightarrow R \longrightarrow \lim_{\gamma \to 1} \mathfrak{s}_+^{\gamma}. \tag{23}$$

Applying the data processing inequality [50], we get

$$I\Big(\lim_{\gamma \to \infty} \mathfrak{s}_+^{\gamma}\, ; \mathfrak{a}_t^{\mathbf{H}}, \dots, \mathfrak{a}_{t+K}^{\mathbf{H}} \ \Big|\ \widetilde{\mathfrak{s}}_t\Big) \le I\big(R; \mathfrak{a}_t^{\mathbf{H}}, \dots, \mathfrak{a}_{t+K}^{\mathbf{H}} \mid \widetilde{\mathfrak{s}}_t\big), \tag{24}$$

from which Eq. (21) follows. $\qquad \square$

**Lemma B.2.** *Suppose we have $k$ logits, denoted by the map $\alpha : \{1 \ldots k\} \to [0,1]$. For any $\beta > 0$, we can construct the (softmax) distribution*

$$p_\beta(i) \propto \exp\big(\beta \alpha(i)\big).$$

*Then,*

$$\mathcal{H}(p_\beta) \geq \log k - \left(\frac{\beta}{e}\right)^2. \tag{25}$$

*Proof.* We lower bound the "worst-case" of the RHS, $\alpha = (1, 0, \ldots, 0)$:

$$
\begin{aligned}
\mathcal{H}(p_\beta) &= \frac{(1-n)\log\big(\frac{1}{k+e^\beta-1}\big)}{k+e^\beta-1} - \frac{e^\beta \log\big(\frac{e^\beta}{k+e^\beta-1}\big)}{k+e^\beta-1} \\
&= \frac{(k+e^\beta-1)\log(k+e^\beta-1) - e^\beta \log(e^\beta)}{k+e^\beta-1} \\
&= \log(k+e^\beta-1) - \frac{e^\beta \log(e^\beta)}{k+e^\beta-1} \\
&\geq \log k - (\beta/e)^2. \tag{26}
\end{aligned}
$$

$\square$

**Lemma B.3.** *For any $t$ and $K \geq 0$,*

$$I(R; \mathfrak{a}_t^{\mathbf{H}}, \ldots, \mathfrak{a}_{t+K}^{\mathbf{H}} \mid \widetilde{\mathfrak{s}}_t) \leq \lim_{\gamma \to 1}\left(\frac{\beta}{e}\mathbb{E}\big[Q_{R,\gamma}^{\pi_{\mathbf{H}}, \pi_{\mathbf{R}}}(s_t, \mathfrak{a}_t^{\mathbf{H}}, \ldots, \mathfrak{a}_{t+K}^{\mathbf{H}})\big]\right)^2. \tag{27}$$

*Proof.* Denote by $\hat{\mathfrak{a}}_t^{\mathbf{H}} \ldots \hat{\mathfrak{a}}_{t+K} \sim \mathrm{Unif}(\mathcal{A}^{\mathbf{H}})$ a sequence of $K$ random actions. From Lemma B.2:

$$
\begin{aligned}
I(R; \mathfrak{a}_t^{\mathbf{H}}, \ldots, \mathfrak{a}_{t+K}^{\mathbf{H}} \mid \widetilde{\mathfrak{s}}_t) &= \mathcal{H}(\mathfrak{a}_t^{\mathbf{H}}, \ldots, \mathfrak{a}_{t+K}^{\mathbf{H}} \mid \widetilde{\mathfrak{s}}_t) - \mathcal{H}(\mathfrak{a}_t^{\mathbf{H}}, \ldots, \mathfrak{a}_{t+K}^{\mathbf{H}} \mid R, \widetilde{\mathfrak{s}}_t) \\
&\leq \log\big(K|\mathcal{A}|\big) - \mathcal{H}\big(\mathfrak{a}_t^{\mathbf{H}}, \ldots, \mathfrak{a}_{t+K}^{\mathbf{H}} \mid R, \widetilde{\mathfrak{s}}_t\big) \\
&\leq \lim_{\gamma \to 1}\left(\frac{\beta}{e}\mathbb{E}\big[Q_{R,\gamma}^{\pi_{\mathbf{H}}, \pi_{\mathbf{R}}}(s_t, \mathfrak{a}_t^{\mathbf{H}}, \ldots, \mathfrak{a}_{t+K}^{\mathbf{H}}) - Q_{R,\gamma}^{\pi_{\mathbf{H}}, \pi_{\mathbf{R}}}(s_t, \hat{\mathfrak{a}}_t^{\mathbf{H}}, \ldots, \hat{\mathfrak{a}}_{t+K}^{\mathbf{H}})\big]\right)^2, \tag{28}
\end{aligned}
$$

where the last inequality follows from Lemma B.2 and $Q_{R,\gamma}^{\pi_{\mathbf{H}}, \pi_{\mathbf{R}}}(\ldots) \leq 1$. We also have

$$0 \leq Q_{R,\gamma}^{\pi_{\mathbf{H}}, \pi_{\mathbf{R}}}(s_t, \hat{\mathfrak{a}}_t^{\mathbf{H}}, \ldots, \hat{\mathfrak{a}}_{t+K}^{\mathbf{H}}) \leq Q_{R,\gamma}^{\pi_{\mathbf{H}}, \pi_{\mathbf{R}}}(s_t, \mathfrak{a}_t^{\mathbf{H}}, \ldots, \mathfrak{a}_{t+K}^{\mathbf{H}}) \leq 1, \tag{29}$$

which lets us conclude from Eq. (28) that

$$I(R; \mathfrak{a}_t^{\mathbf{H}}, \ldots, \mathfrak{a}_{t+K}^{\mathbf{H}} \mid \widetilde{\mathfrak{s}}_t) \leq \left(\frac{\beta}{e}\mathbb{E}\big[Q_{R,\gamma}^{\pi_{\mathbf{H}}, \pi_{\mathbf{R}}}(s_t, \mathfrak{a}_t^{\mathbf{H}}, \ldots, \mathfrak{a}_{t+K}^{\mathbf{H}})\big]\right)^2. \tag{Eq. 27}$$

$\square$

We can now prove Theorem 3.1 directly by combining Lemmas B.1 and B.3.

*Proof of Theorem 3.1.* Simplifying the limit in Eq. (9), we get

$$
\begin{aligned}
\liminf_{\gamma \to 1} \mathcal{E}_\gamma(\pi_{\mathbf{H}}, \pi_{\mathbf{R}}) &\leq \liminf_{\gamma \to 1}\left(\sum_{t=0}^\infty \gamma^t I(\mathfrak{s}_+^\gamma; \mathfrak{a}_t^{\mathbf{H}} \mid \widetilde{\mathfrak{s}}_t)\right) \\
&\leq \liminf_{\gamma \to 1} I(\mathfrak{s}_+^\gamma; \mathfrak{a}_t^{\mathbf{H}}, \ldots, \mathfrak{a}_{t+K}^{\mathbf{H}} \mid \widetilde{\mathfrak{s}}_t) && \text{(chain rule)} \\
&\leq I(R; \mathfrak{a}_t^{\mathbf{H}}, \ldots, \mathfrak{a}_{t+K}^{\mathbf{H}} \mid \widetilde{\mathfrak{s}}_t) && \text{(Lemma B.1)} \\
&\leq \lim_{\gamma \to 1}\left(\frac{\beta}{e}\mathbb{E}\big[Q_{R,\gamma}^{\pi_{\mathbf{H}}, \pi_{\mathbf{R}}}(s_t, \mathfrak{a}_t^{\mathbf{H}}, \ldots, \mathfrak{a}_{t+K}^{\mathbf{H}})\big]\right)^2 && \text{(Lemma B.3)} \\
&\leq \lim_{\gamma \to 1}\left(\frac{\beta \mathcal{J}_{\pi_{\mathbf{R}}}^\gamma(\pi_{\mathbf{H}})}{e}\right)^2. \tag{30}
\end{aligned}
$$

It follows that for sufficiently large $\gamma$,

$$\mathcal{E}_\gamma(\pi_{\mathbf{H}}, \pi_{\mathbf{R}})^{1/2} \leq (\beta/e)\, \mathcal{J}_{\pi_{\mathbf{R}}}^\gamma(\pi_{\mathbf{H}}). \tag{Eq. 9}$$

$\square$

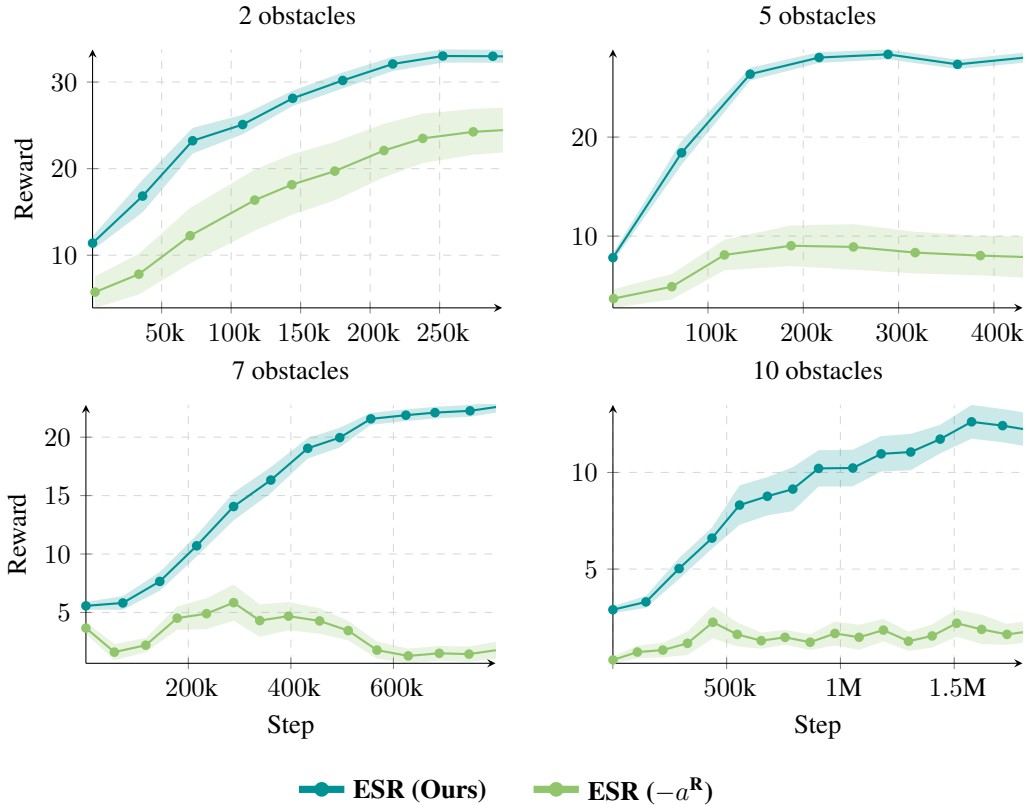

Figure 6: We evaluate our method with and without conditioning on the robot action $a^{\mathbf{R}}$. Conditioning aids learning significantly, which we theorize is because it removes uncertainty in the classification.

## C Additional Ablations and Qualitative Results

In this section we evaluate additional ablations and qualitative results for the ESR method.

### C.1 Learning Representations without the Robot Action

In our estimation of empowerment (Eq. 12) we supply the robot action $a^{\mathbf{R}}$ when learning both $\phi$ and $\phi$, however, the theoretical empowerment formulation in Section 3.3 does not require it.

To evaluate the impact of including $a^{\mathbf{R}}$, we run an additional ablation without it on the gridworld environment, shown in Fig. 6. This ablation shows that the inclusion of $a^{\mathbf{R}}$ is most impactful in more challenging (higher number of boxes) environments. We hypothesize that conditioning the representations on the robot action reduces the noise in the mutual information estimation, and also reduces the difficulty of classifying true future states.

## D Greedy Empowerment Policy

All of our experiments have used Soft-Q learning to learn a policy from the empowerment estimation. Here, we additionally study a greedy empowerment policy which takes the most empowering action at each step. We model this by setting the q-learning gamma to 0 to fully discount future rewards.

Results for this ablation are shown in Fig. 7. Unsurprisingly, the greedy optimization vastly underperforms the policy with $\gamma = 0.9$.

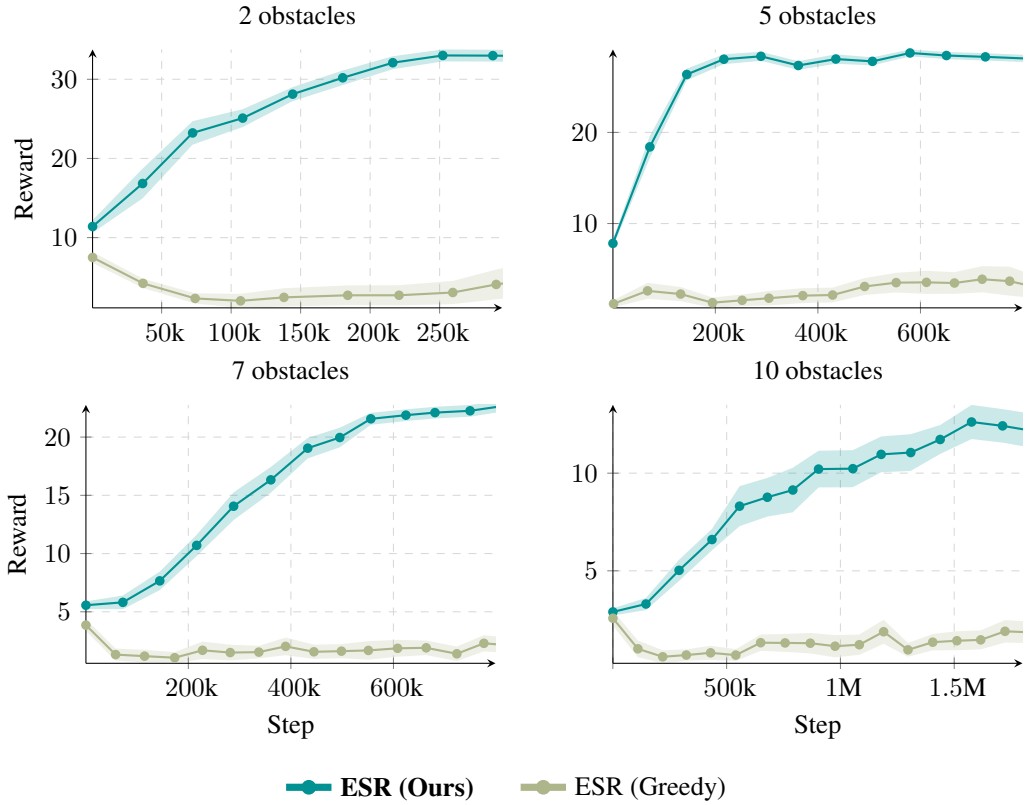

Figure 7: We compare a greedy policy ($\gamma = 0$) against our standard policy ($\gamma = 0.9$).

## D.1 ESR Training Example

In Fig. 8, we show the mutual information during training of the ESR agent in the gridworld environment with 5 obstacles. he mutual information quickly becomes positive and remains so throughout training. As long as the mutual information is positive, the classifier is able to reward the agent for taking actions that empower the human.

## E Simplifying the Objective

The reward function in Eq. (13) is itself a random variable because it depends on future states $g$. This subsection describes how this randomness can be removed. To do this, we follow prior work [64, 65] in arguing that the learned representations $\psi(g)$ follow a Gaussian distribution:

**Assumption E.1** (Based on Wang and Isola [64]). *The representations of future states $\psi(g)$ learned by contrastive learning have a marginal distribution that is Gaussian:*

$$\mathrm{P}(\psi) = \int \mathrm{P}(g)\delta(\psi = \psi(g))\,\mathrm{d}g \overset{d}{=} \mathcal{N}(0, I). \tag{31}$$

With this assumption, we can remove the random sampling of $g$ from the reward function. We start by noting that the learned representations tell us the *relative* likelihood of seeing a future state Eq. (12). Assumption E.1 will allow us to convert these relative likelihoods into likelihoods.

$$
\begin{aligned}
\mathbb{E}_{\mathrm{P}(\mathfrak{s}^+|s,a^{\mathbf{R}},a^{\mathbf{H}})}[r(s,a^{\mathbf{R}})] &= \mathbb{E}_{\mathrm{P}(\mathfrak{s}^+)}\left[\frac{\mathrm{P}(\mathfrak{s}^+|s,a^{\mathbf{R}},a^{\mathbf{H}})}{\mathrm{P}(\mathfrak{s}^+)}r(s,a^{\mathbf{R}})\right] \\
&= \mathbb{E}_{\mathrm{P}(\mathfrak{s}^+)}\left[C_1 e^{\phi(s,a^{\mathbf{R}},a^{\mathbf{H}})^T\phi(\mathfrak{s}^+)}r(s,a^{\mathbf{R}})\right] \\
&= C_1 \mathbb{E}_{\psi\sim\mathrm{P}(\phi(\mathfrak{s}^+))}\left[e^{\phi(s,a^{\mathbf{R}},a^{\mathbf{H}})^T\psi}(\phi(s,a^{\mathbf{R}},a^{\mathbf{H}}) - \phi(s,a^{\mathbf{R}}))^T\psi\right]
\end{aligned}
$$

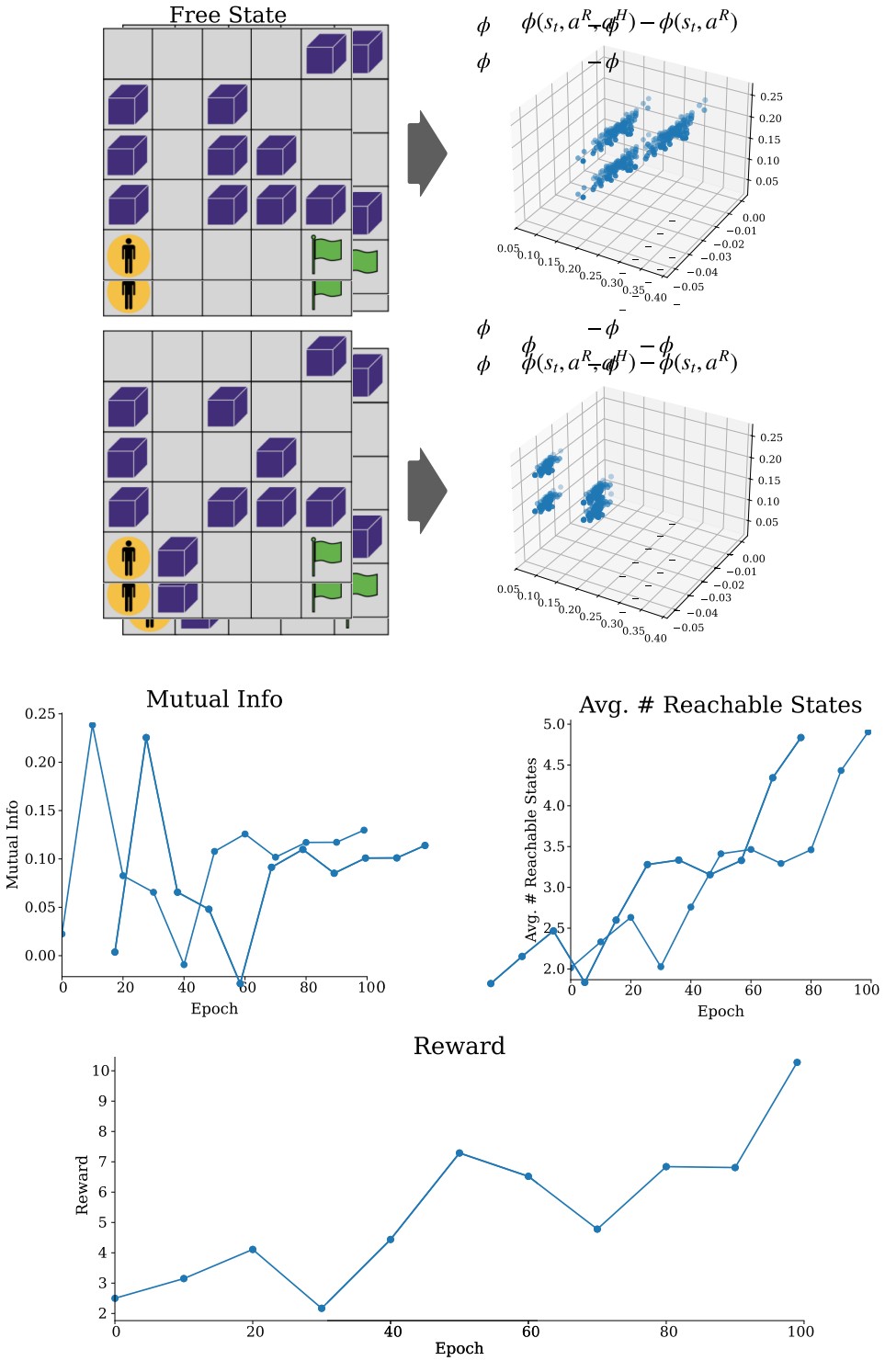

Figure 8: Visualizing training empowerment in a 5x5 Gridworld with 10 obstacles. Our empowerment objective maximizes the influence of the human's actions on the future state, preferring the state where the human can reach the goal to the trapped state. This corresponds to maximizing the volume of the state marginal polytope, which is proportional to the number of states that the human can reach from their current position. To visualize the representations, we set the latent dimension to 3 instead of 100.

$$= C_1 \big(\phi(s, a^{\mathbf{R}}, a^{\mathbf{H}}) - \phi(s, a^{\mathbf{R}})\big)^T$$

$$\int \frac{1}{(2\pi)^{d/2}} e^{-\frac{1}{2}\|\psi\|_2^2 + \phi(s, a^{\mathbf{R}}, a^{\mathbf{H}})^T \psi} \psi \, \mathrm{d}\psi$$

$$= C_1 \big(\phi(s, a^{\mathbf{R}}, a^{\mathbf{H}}) - \phi(s, a^{\mathbf{R}})\big)^T e^{\frac{1}{2}\|\phi(s, a^{\mathbf{R}}, a^{\mathbf{H}})\|_2^2}$$

$$\int \frac{1}{(2\pi)^{d/2}} e^{-\frac{1}{2}\|\psi\|_2^2 + \phi(s, a^{\mathbf{R}}, a^{\mathbf{H}})^T \psi - \frac{1}{2}\|\phi(s, a^{\mathbf{R}}, a^{H})\|_2^2} \psi \, \mathrm{d}\psi$$

$$= C_1 \big(\phi(s, a^{\mathbf{R}}, a^{\mathbf{H}}) - \phi(s, a^{\mathbf{R}})\big)^T$$

$$e^{\frac{1}{2}\|\phi(s, a^{\mathbf{R}}, a^{\mathbf{H}})\|_2^2} \mathbb{E}_{\psi \sim \mathcal{N}(\mu = \phi(s, a^{\mathbf{R}}, a^{\mathbf{H}}), \Sigma = I)} \big[\psi\big]$$

$$= C_1 e^{\frac{1}{2}\|\phi(s, a^{\mathbf{R}}, a^{\mathbf{H}})\|_2^2} \big(\phi(s, a^{\mathbf{R}}, a^{\mathbf{H}}) - \phi(s, a^{\mathbf{R}})\big)^T \phi(s, a^{\mathbf{R}}, a^{\mathbf{H}}). \quad (32)$$

This simplification may be attractive in in cases where the computed empowerment bonuses have high variance, or when the empowerment horizon is large (i.e., $\gamma \to 1$, as in Section 3.3). Empirically, we found this version of the objective to be less effective in practice due to the additional representation structure required by Assumption E.1.

