# OpenReview forum: "Learning to Assist Humans without Inferring Rewards"
_NeurIPS.cc/2024/Conference — NeurIPS 2024 poster_

### Official Review · Reviewer_KT7u · 2024-07-08

**Soundness:** 2
**Presentation:** 3
**Contribution:** 2
**Rating:** 4
**Confidence:** 4

**Summary:**

This paper presents Empowerment via Successor Representations or ESR, a technique that builds an assistive agent that maximizes the human collaborator's ability to influence the world. The key motivation the authors provide for building an assistive agent that seeks to empower the human rather than explicitly provide support by inferring the human collaborator's reward function is that inferring such a reward function can be challenging. The authors present a formulation for empowerment and connect empowerment maximization to reward maximization in Section 3. The authors then present an implicit approach to estimate empowerment via several learned representations and present loss functions for how these representations can be inferred and utilized. Finally, the authors present a set of experiments displaying that this technique can outperform a related prior work (AvE) as well as a random baseline.

**Strengths:**

Strengths
+ The presented formulation is very interesting and shows promise. To the best of my knowledge, the utilization of such representations to estimate empowerment is new.
+ Generally, the results are impressive. It is clear that ESR outperforms the prior technique by large margins in the chosen domains.

**Weaknesses:**

Weaknesses
- Paper's claims should be better grounded:
1. For the proof in Section 3.3, the conclusion is that an assistive agent maximizing the mutual information between a future state and a chosen skill by the human minimizes the worst-case regret incurred by the human. Is this only true for a two-agent scenario and full observability? Does the proof have any assumptions for the robot and human maintaining the same representation and action space?
2. The paper mentions that inferring a human's reward function can be error-prone and challenging. However, I would think inference of several representations (especially online), could lead to similar outcomes (poor consequences etc.) It isn't clear why inferring these approximate representations and seeking to maximize empowerment would work better than reward inference. I'm specifically thinking about the cases mentioned in the introduction, where humans have changing preferences, act suboptimally, etc.
3. The term high-dimensional is used in the introduction but the chosen domains seem to be relatively low-dimensional compared to other collaborative testbeds encountered in robotics (see [1]) and human-AI collaboration (see [2]). Could you clarify this term?
- Evaluation is limited:
1. As this work aims to develop assistive agents that work with humans, it should compare with actual humans in a user study to validate the synthetic findings.
2. The evaluation is only conducted with respect to one baseline (AvE) and a random agent. The author's key rationale behind this technique was assisting humans without inferring reward functions, so it would be beneficial to test against a framework that actively infers the user's reward function.
- Some claims are not well-explained.
1. The introduction notes that AvE does not scale to high-dimensional settings but does not make it clear why this framework failed to scale.
2. The authors show and mention AvE underperforms a random agent but does not mention why. Could you provide further details? Also, could the authors comment on why ESR was able to outperform AvE by such a wide margin and note any qualitative findings about the learned collaborative behavior?


[1] Liu, Puze, et al. "Safe reinforcement learning of dynamic high-dimensional robotic tasks: navigation, manipulation, interaction." 2023 IEEE International Conference on Robotics and Automation (ICRA). IEEE, 2023.

[2] Paleja, Rohan, et al. "The utility of explainable ai in ad hoc human-machine teaming." Advances in neural information processing systems 34 (2021): 610-623.

**Questions:**

Other Questions/Recommendations:
- Could you please comment on the weaknesses above?
- Could you comment on the feasibility of this approach with actual humans? How many samples would be needed to infer semi-accurate representations and create agents that could collaborate well? Further, if the answer is learning these representation a priori, could you comment on how this framework would work in cases where you do not have access to a high-fidelity simulator.
- There is some notation that is unclear and grammar issues/typos. I would recommend checking to ensure that all variables are defined, which will help improve clarity.
- Figure Descriptions could be improved

**Limitations:**

Yes.

---

> ### Author Rebuttal · Authors · 2024-08-07
>
> Dear Reviewer,
>
> Thank you for the detailed review, and for all the suggestions for improving the paper.
> Based on the reviewer's feedback, we have run experiments with additional baselines and clarified several parts of the writing.
> **Together with the discussion below, does this fully address the reviewer's concerns with the paper?** We look forward to continuing the discussion!
>
> Kind regards,
>
> The authors
>
> > It isn't clear why inferring these approximate representations and seeking to maximize empowerment would work better than reward inference.
> > it would be beneficial to test against a framework that actively infers the user's reward function.
>
> To answer this question, we ran an additional experiment comparing our method (which estimates representations) to a reward learning approach that uses IQ-Learn (Garg et al., 2021) as an IRL algorithm to infer human rewards and then use these to train the assistant, as in the CiRL (Hadfield-Menell et al., 2016) framework. The results (rebuttal PDF Figs. 1 and 3) show that our approach outperforms this baseline.
>
> > Is this only true for a two-agent scenario and full observability?
>
> Yes. We have revised the paper to add these assumptions. We believe that handling more than two agents should be feasible (in future work) but handling partial observability may be more challenging. Under certain assumptions (e.g., state-independent noise in observing human actions or bounded divergence between the agent’s belief and the true environment) we can reason about these challenges. But without any assumptions pathological MDPs can be constructed.
>
> > Does the proof have any assumptions for the robot and human maintaining the same representation and action space?
>
> No, they do not need to have the same representations or action space. Our implementation re-uses one representation ($\psi(g)$) purely for computationally efficiency, but this can be removed if an algorithm user wanted to use entirely disjoint representations.
>
> > Clarify the meaning of high-dimensional
>
> Our states for both environments are images: 5x5 with 3 feature channels for the obstacle gridworld and 4x5/5x5 with 26 feature channels for the Overcooked environments.
>
>
> > Why does AvE perform so poorly on these experiments?
>
> The key limitation of AvE is it tries to approximate the empowerment quantity which they define as $\max_\pi I(s^+;a_t\mid s_t)$ through random rollouts. This is completely intractable in high-dimensional environments (as our larger experiments show), where random rollouts don’t produce meaningful behaviors.
>
> Additionally, since the rollouts select actions uniformly at random, even in the limit of infinite computation, the AvE objective doesn’t actually compute empowerment with respect to meaningful human behaviors (either under their definition or our definition). Intuitively, imagine a human performing some task in a room with thin walls. The AvE objective would incentivize the robot knocking down the walls to maximize the human’s ability to leave the house, even though they have no desire to do so. Our ESR objective would help them with the task at hand.
>
> > why ESR was able to outperform AvE by such a wide margin and note any qualitative findings about the learned collaborative behavior?
>
> For the reasons noted above, AvE is unable to perform well on the more complex tasks studied, while ESR can scalably empower the human. Qualitatively, in the obstacle environment ESR learns to remove obstacles along the path the human is moving and in the Overcooked setting it learns collaborative behaviors like moving the plate to places the human can reach and placing onions in the pot. We will revise the paper with sketches of some of these behaviors.
>
> > Could you comment on the feasibility of this approach with actual humans?
>
> This paper is primarily mathematical and algorithmic and nature, and so we'll provide an mathematical/algorithmic answer: Eq. (6) says that we can measure the degree of (minimax) compatibility via the mutual information. We have run an additional experiment to plot this mutual information throughout training (see Fix X in rebuttal PDF). Visualizing the learned agent, we see that the agent does indeed become more helpful as this mutual information increases.
>
> Of course, the gold standard in human-AI interaction is human user studies, which are beyond the scope of this mathematical/algorithmic paper.
>
>
> > checking to ensure that all variables are defined, which will help improve clarity.
>
> Thanks for the suggestion – we have done this with our local working copy of the paper.
>
> > Figure Descriptions could be improved
>
> We have clarified the figure descriptions, incorporating the feedback from the other reviewers as well.

---

> ### Author Response · Authors · 2024-08-07
> **References**
>
> Du, Yuqing, Stas Tiomkin, Emre Kiciman, Daniel Polani, Pieter Abbeel, and Anca Dragan. 2020. “AvE: Assistance via Empowerment.” Pp. 4560–71 in _Advances in Neural Information Processing Systems_. Vol. 33. Curran Associates, Inc.
>
> Garg, Divyansh, Shuvam Chakraborty, Chris Cundy, Jiaming Song, and Stefano Ermon. 2021. “IQ-Learn: Inverse Soft-Q Learning for Imitation.” Pp. 4028–39 in _Advances in Neural Information Processing Systems_. Vol. 34. Curran Associates, Inc.
>
> Hadfield-Menell, Dylan, Stuart J. Russell, Pieter Abbeel, and Anca Dragan. 2016. “Cooperative Inverse Reinforcement Learning.” _Advances in Neural Information Processing Systems_ 29.

---

> > ### Comment · Reviewer_KT7u · 2024-08-11
> >
> > Dear Authors,
> >
> > Thank you for your response. I appreciate the new results, the clarification regarding assumptions, and additional information about AvE. As a lack of human assessment and compatibility with actual humans was noted by several reviewers, this should be emphasized in the limitations.
> >
> > After reading all the reviews and their respective replies, I have decided to increase my score.

---

> ### Author Response · Authors · 2024-08-11
> **Response**
>
> Thank you for your response! We have revised the paper to mention this limitation in the conclusion. We would like to note that numerous papers published at NeurIPS and similar venues also aim to establish the algorithmic and theoretical foundations for human-AI interaction and alignment before proceeding with human studies (Ammanabrolu et al., 2022; Chan et al., 2019; Hadfield-Menell et al., 2016, 2017; He et al., 2023; Ngo et al., 2024; A. Pan et al., 2022; M. Pan et al., 2024; Zhuang & Hadfield-Menell, 2020). We will adjust the introduction to indicate that our paper extends this line of work, providing:
> 1. an algorithmic and theoretical framework for creating aligned AI agents without the machinery of inferring human values (such as in the CIRL framework of Hadfield-Menell et al., (2016, 2017)), and
> 2. a proof-of-concept showing scalable contrastive estimators enable improved unsupervised assistance in synthetic benchmarks from past work.
>
>
> ### References
>
> Ammanabrolu, Prithviraj, Liwei Jiang, Maarten Sap, Hannaneh Hajishirzi, and Yejin Choi. 2022. “Aligning to Social Norms and Values in Interactive Narratives.” in _NAACL-HLT_. arXiv.
>
> Chan, Lawrence, Dylan Hadfield-Menell, Siddhartha Srinivasa, and Anca Dragan. 2019. “The Assistive Multi-Armed Bandit.” in _ACM/IEEE International Conference on Human-Robot Interaction_. arXiv.
>
> Hadfield-Menell, Dylan, Smitha Milli, Pieter Abbeel, Stuart J. Russell, and Anca Dragan. 2017. “Inverse Reward Design.” _Advances in Neural Information Processing Systems_ 30.
>
> Hadfield-Menell, Dylan, Stuart J. Russell, Pieter Abbeel, and Anca Dragan. 2016. “Cooperative Inverse Reinforcement Learning.” _Advances in Neural Information Processing Systems_ 29.
>
> He, Jerry Zhi-Yang, Daniel S. Brown, Zackory Erickson, and Anca Dragan. 2023. “Quantifying Assistive Robustness via the Natural-Adversarial Frontier.” Pp. 1865–86 in _Proceedings of The 7th Conference on Robot Learning_. PMLR.
>
> Ngo, Richard, Lawrence Chan, and Sören Mindermann. 2024. “The Alignment Problem from a Deep Learning Perspective.” in _The twelfth international conference on learning representations_.
>
> Pan, Alexander, Kush Bhatia, and Jacob Steinhardt. 2022. “The Effects of Reward Misspecification: Mapping and Mitigating Misaligned Models.” in _International Conference on Learning Representations_. arXiv.
>
> Pan, Michelle, Mariah Schrum, Vivek Myers, Erdem Bıyık, and Anca Dragan. 2024. “Coprocessor Actor Critic: A Model-Based Reinforcement Learning Approach For Adaptive Brain Stimulation.” in _International Conference on Machine Learning_.
>
> Zhuang, Simon, and Dylan Hadfield-Menell. 2020. “Consequences of Misaligned AI.” Pp. 15763–73 in _Advances in Neural Information Processing Systems_. Vol. 33. Curran Associates, Inc.

---

### Official Review · Reviewer_MFVx · 2024-07-13

**Soundness:** 3
**Presentation:** 2
**Contribution:** 3
**Rating:** 5
**Confidence:** 4

**Summary:**

This paper introduces a method for assistance via empowerment based on contrastive successor representations, while also introducing a number of theoretical results about the relationship between assistive empowerment (understood as maximizing the mutual information between human actions and future states) and assistive reward maximization in information geometric terms. The proposed method, empowerment with successor representations (ESR) is shown in experiments to outperform a baseline method, AvE, which estimates empowerment using Monte Carlo rollouts. The experiments show that ESR scales to more complex problems than AvE, including environments with image based observations.

**Strengths:**

This paper introduces a new method, ESR, for training agents to assist users by maximizing empowerment, which does not require the agent to infer human goals or preferences (as in methods based on inverse planning or inverse reinforcement learning). Unlike previous methods for assistance via empowerment, which make use of Monte Carlo rollouts to estimate variance, ESR's method for estimating empowerment is more scalable, drawing upon ideas in contrastive representation learning to effectively estimate the mutual information between human actions and future states. This allows ESR to be applied to more complex problems, increasing the applicability of assistance via empowerment to more contexts. The underlying ideas behind this method are interesting, and the method appears to be effective (at least with respect to the AvE baseline). As such, I believe the paper will be of some interest to others working on AI assistance and human-AI alignment.

**Weaknesses:**

While the ideas behind this paper are interesting, and the method seems to work reasonably well, the presentation of the framework could be significantly improved. The empirical evaluations could also be made more rigorous by comparing against non empowerment-based baselines.

**Presentation**

This paper was hard to understand on its own, even after reading the Appendix. I had to read Eysenbach (2021) to really understand the theoretical results, and van den Oord (2019) to understand how the proposed method worked. I think a well-presented paper wouldn't have had that issue.

In the information geometry section, important quantities such as the skill variable $z$ and state occupancy measure $\rho(s)$ are not properly defined. This makes it hard to understand all the different state occupancy measures $\rho(z)$, $\rho^+(z)$, $\rho^*(z)$ that are introduced, and what exactly the prior over states means (or why the human would be able to choose this prior). As a result, it's hard to evaluate the soundness of the arguments (without, e.g., reading Eysenbach (2021)), or whether the assumptions about the human policy underlying Lemma 2 are reasonable ones. It's also not explained how the mutual information $I(s^+; z)$. between future states $s^+$ and human skills $z$ is related to the definition of empowerment in Equation (1), which involves the (conditional) mutual information $I(s^+; a^H_t | s_t)$ between future states and human actions $a^H_t$. Since this is not explained, it's hard to see how Section 3.2 and 3.3 are relevant for actually maximizing empowerment as it's defined in Section 3.1.

In section 4, where the ESR algorithm is introduced, it's also not explained how or why the contrastive representation learning objective in Eq. (7) should lead to right density ratios to be estimated upon training convergence. One has to read van den Oord (2019) to understand why, so this section is not really accessible to readers not already familiar with contrastive learning methods. Notation is also not clearly defined -- for example, why does $\phi$ switch from being a 3-argument function in Line 191 to becoming 2 argument function in Line 198? I have the inkling that this has to do with marginalizing out the assistant's action $a^R$, but this is not explained. I'm also confused why $a^R$ needs to be part of the successor representation at all. By including $a^R$, won't the resulting mutual information you're estimating end up being $I(s^+; a^H_t | s_t, a^R_t )$, which is conditional on *both* the current state $s_t$ and the assistant's action $a^R_t$?

Perhaps one reason why this paper ends up being hard to follow is because it's trying to do too much by both introducing the results in Sections 3.2-3.3, while also introducing a (seemingly unrelated) method for empowerment maximization in Section 4. As it stands, these two parts of the paper feel quite disjoint to me, and it's not obvious how they form a cohesive whole. It might have been better to focus on just one or the other -- e.g. on just explaining and carefully justifying the method in Section 4 -- so that everything is more understandable and cohesive.

**Evaluation**

Even though one of the main selling points of assistance via empowerment is that it does not require performing goal inference (which can be hard to scale when the goal space is large), the experiments do not compare ESR against any goal inference methods. This is in contrast to the original AvE paper by Du et al (2020), which does conduct fairly thorough comparisons against goal inference baselines. As a result, it's hard to evaluate exactly how valuable ESR is, and whether one should prefer it over goal inference as an assistance method. This should be addressed in future versions of the paper.

**Questions:**

POST-REBUTTAL: In light of the new experimental results comparing ESR to a reward inference baseline, I have raised my score to a 5. However, there remain substantial improvements to the presentation and theory-to-algorithm connection that should be made.

===

Questions about empowerment definition:

- Equation 1: What is the expectation in Eq (1) taken over? Is it over $\pi_H , \pi_R$? Is $s^+$ sampled $K$ steps into the future starting from each $t$? Or starting from $t = 0$?

Questions about information geometry:

- Figure 2: What exactly is the 'center of the polytope" here?
- Lemma 1 and Lemma 2: What exactly is the skill $z$, and how is it related to human actions $a^H$?
- Relatedly, how is the mutual information $I(s^+; z)$ related to $I(s^+; a^H_t | s_t)$?
- In Lemma 2, what does it mean for a human to "adapt to a reward function"? If I'm understanding Eysenbach et al (2021) correctly, is the idea here that the human is modeled as starting from prior state distribution $\rho(s)$, then adapting to $\rho*(s)$ by learning to better optimize for the reward function (by learning new skills)?
- In Lemma 2, I can see that the assistant maximizing the value of $I(s^+; z)$ leads to lower (regularized) regret for the human. But does maximizing the discounted sum of $I(s^+; a^H_t | s_t)$ also lead to lower regret for the human? It seems like this result isn't shown, and so Lemma 2 doesn't actually apply to the notion of empowerment used in the paper.
- Line 159: "We can view our objective as a generalization of the assistance problem beyond the CIRL setting" --- I would be careful about making this claim. The assistance game setting is a (cooperative) Markov game, and the solution concepts in that setting are either optimal policies (Hadfield-Menell et al, 2016) or pragmatic-pedagogical equilibria (Fisac et al, 2017). In contrast, Lemma 2 only shows that maximizing mutual information corresponds to minimizing regularized regret --- which is not the same as finding the optimal joint policy, or finding a pragmatic-pedagogical equilibrium.

Questions about contrastive representation learning:

- Line 197: What is $g$? A future state? Why not use $s^+$ as before?
- Line 198: Why do $\phi$ and $\phi'$ each suddenly lose one argument?
- Line 201: Please provide some derivation or explanation as to why these representations would encode the stated probability ratios upon convergence.
- Equations 8 and 9: Why are the conditional probabilities not conditioned on $a^R_t$?

Questions about experiments:

- What embedding functions or neural network did you use to learn the successor features in each benchmark?
- How does ESR compare against goal inference baselines, e.g. those used in Du et al (2020) or Laidlaw et al (2024)?

Minor Comments:

- There are a number of typos here and there ("gradefully", "belore") which should be fixed.
- Line 256: "While much of the amazing work" --- too subjective for an academic paper.

**Limitations:**

The authors adequately discuss the technical limitations of their approach. They also have appropriately noted the risks of focusing solely on empowerment for assistance, as this might empower actors who already unjustly have too much power.

---

> ### Author Rebuttal · Authors · 2024-08-07
>
> Dear Reviewer,
>
> Thank you for the detailed review, and for the suggestions for improving the paper. It seems like the reviewer's main concern is about baselines and presentation. We have attempted to address these concerns by adding XX additional baselines, and by significantly revising the paper (incorporating the reviewer's suggestions). **Together with the discussion below, does this fully address the reviewer's concerns?** If not, we look forward to continuing the discussion!
>
> Kind regards,
>
> The authors.
>
> > How does ESR compare against goal inference baselines, e.g. those used in Du et al (2020) or Laidlaw et al (2024)?
>
> We have added goal inference baselines for the obstacle environment, with our ESR method achieving the best performance (see attached PDF). In contrast with Du et al (2020), which assumes a model-based, almost “oracle” goal inference, we implement a model-free baseline for fair comparison. We have revised the paper to describe this.
>
> In the Overcooked environments, there is no clear notion of a goal (the environment returns roughly to the same state it started in after the soup is made and delivered).  An advantage of our method over goal inference is that it is well-defined in such settings.
>
> > Equation 1: What is the expectation in Eq (1) taken over? Is it over $\pi_H, \pi_R$ ? Is $s^{+}$sampled $K$ steps into the future starting from each $t$ ? Or starting from $t=0$?
>
> This expectation is taken over a trajectory $(s_0, s_1, \cdots)$ sampled when the human ($\pi_H$) and robot ($\pi_R$) interact in the environment. Thus, the random variable $s_t$ corresponds to time step $t$ in an episode. Random variable $s^+$ is sampled from the discounted state occupancy measure conditioned on $s_t$; in other words, the sampling procedure can be written as $K \sim Geom(1 - \gamma)$ and setting $s^+ = s_{t+K}$. We have revised the paper to clarify this.
>
> > important quantities such as the skill variable z and state occupancy measure ρ(s) are not properly defined.
>
> We will revise the paper to more clearly connect the notation used for the analysis to the main text.
>
> > how the mutual information  I(s+;z) . between future states s+ and human skills z is related to the definition of empowerment in Equation (1), which involves the (conditional) mutual information I(s+;atH|st) between futurestates and human actions atH In section 4, where the ESR algorithm is introduced, it's also not explained how or why the contrastive representation learning objective in Eq. (7) should lead to right density ratios to be estimated upon training convergence. One has to read van den Oord (2019) to understand why, so this section is not really accessible to readers not already familiar with contrastive learning methods.
>
> Thank you for pointing this out! We agree that it is difficult to understand the mathematical details without this background. In our revision, we will add a derivation of the solution to our symmetrized infoNCE objective in Appendix D, which follows directly from the analysis by Poole et al. (2019) when adapted to the symmetric version of the loss as in Radford et al. (2021) and Eysenbach et al. (2024).
>
> > why aR needs to be part of the successor representation at all. By including aR, won't the resulting mutual information you're estimating end up being I(s+;atH|st,atR), which is conditional on both the current state st and the assistant's action atR ?
>
> We found conditioning on a_R as well helped stabilize training. A challenge with training an algorithm like ESR is that as $\pi_R$ changes during training, it affects the successor features and the computed empowerment reward. By conditioning the representation on the current policy’s actions, we improve the ability of the empowerment bonus to “keep up” with the policy during training. We will revise the main text to indicate this more clearly.
>
> > Lemma 1 and Lemma 2: What exactly is the skill $z$, and how is it related to human actions $a^H$ ?
>
> A ``skill'' is purely a mathematical construction used in our analysis. The analysis focuses on the distribution over states visited by the human and the robot. Our analysis represents the robot's distribution over states $\rho(s)$ as a mixture, $\rho(s) = \sum_z \rho(s \miz s) p(z)$. We have revised the paper to clarify this.
>
> > Relatedly, how is the mutual information $I(s^+; z)$ related to $I(s^+; a_t^H \mid s_t)$?
>
> All instances of random variable $z$ should be replaced by $a^H$; we apologize for any confusion caused by this typo, which was caused by different prior work using different notation. Our theoretical analysis looks at the mutual information $I(s^+; a_t^H)$, which measures the ability of the human to effect change in their environment. Our practical method says that we should maximize this objective at all states: $I(s^+; a_t^H \mid s_t)$ looks at the ability to effect change starting at state $s_t$, and our practical algorithm aims to maximize this objective over all visited states $s_t$ (see Eq. 1).
>
> > In Lemma 2, what does it mean for a human to "adapt to a reward function"? If I'm understanding Eysenbach et al (2021) correctly, is the idea here that the human is modeled as starting from prior state distribution $\rho(s)$, then adapting to $\rho *(s)$ by learning to better optimize for the reward function (by learning new skills)?
>
> Yes, that is correct. In the context of this paper, we look at a restricted set of skills: those defined by (open-loop) sequences of actions, $a_t^H$. This is why we use $a_t^H$ in our mutual information objective, rather than the letter $z$ used in (Eysenbach et al 2021).

---

> ### Author Response · Authors · 2024-08-07
> **Rebuttal (cont.)**
>
> > In Lemma 2, I can see that the assistant maximizing the value of $I\left(s^{+} ; z\right)$ leads to lower (regularized) regret for the human. But does maximizing the discounted sum of $I\left(s^{+} ; a_t^H \mid s_t\right)$ also lead to lower regret for the human? It seems like this result isn't shown, and so Lemma 2 doesn't actually apply to the notion of empowerment used in the paper.
>
> This is an excellent point, one that we didn't realize in our original submission. There remains an important connection between Lemma 2 and the notion of empowerment in the paper (details below), but it is not quite as strong as claimed in the original paper. We will revise the paper accordingly.
>
> The expected value in Lemma 2 corresponds to a summation $\sum_{t=0}^\infty \gamma^t x_t$, where $x_t = \log p(s^+ = s_t \mid a^H)$. The method used in the practical algorithm (Eq. 1) corresponds to a double summation: $\sum_{t=0}^\infty \gamma^t \sum_{i=0}^\infty \gamma^i x_i$. This corresponds to doing RL with a discount factor that is not the usual $\gamma^t$, but rather is $(t+1) \gamma^t$. In summary: yes, the theory differs from the practical algorithm, but the difference corresponds to a different choice of discounting function.
>
> > Line 159: "We can view our objective as a generalization of the assistance problem beyond the CIRL setting" --- I would be careful about making this claim. The assistance game setting is a (cooperative) Markov game, and the solution concepts in that setting are either optimal policies (Hadfield-Menell et al, 2016) or pragmatic-pedagogical equilibria (Fisac et al, 2017). In contrast, Lemma 2 only shows that maximizing mutual information corresponds to minimizing regularized regret --- which is not the same as finding the optimal joint policy, or finding a pragmatic pedagogical equilibrium.
>
> Thank you for raising this point – we will incorporate this into the discussion, clarifying that there are both similarities and differences between our problem formulation and CIRL.
>
>
> > What embedding functions or neural network did you use to learn the successor features in each benchmark?
>
> In the obstacle grid environment, we used a network with 2 convolutional and 2 fully connected layers and SiLU activations. In Overcooked, we adapted the policy architecture from past work (Carroll et al., 2020), using a 3-layer MLP with tanh activations. We will revise the appendix to clearly describe this.
>
> > Figure 2: What exactly is the 'center of the polytope" here?
>
> We have revised the figure to clarify that we are referring to a Barycenter, defined with a KL divergence. In other words, the center is the point that has a minimum KL divergence from the maximally distant point inside the polytope.
>
> > Line 197: What is $g$ ? A future state? Why not use $s^{+}$ as before?
>
> Yes, this is a typo. We have replaced $g$ with $s^+$ here.
>
> > Line 198: Why do $\phi$ and $\phi'$ each suddenly lose one argument?
>
> This is a typo. This line should read: one that aligns $\psi(s, a^R, a^H) \leftrightarrow \psi(s^+)$ and one that aligns $\psi(s, a^R) \leftrightarrow \psi(s^+)$
>
> > Line 201: Please provide some derivation or explanation as to why these representations would encode the stated probability ratios upon convergence.
>
> The citation "[24]" contains a proof of this statement (Poole et al., 2019). We will clarify this in the text. In our revision, we will add a derivation of the solution to our symmetrized infoNCE objective in Appendix D, which follows directly from the analysis by Poole et al. (2019) when adapted to the symmetric version of the loss as in Radford et al. (2021) and Eysenbach et al. (2024).
>
>
> > Equations 8 and 9: Why are the conditional probabilities not conditioned on  $a_t^R$ ?
>
> Thanks for catching this typo – the probabilities in the numerator on the right hand side of both equation should also be conditioned on $a^R$.
>
> > There are a number of typos here and there ("gradefully", "belore") which should be fixed.
>
> We have fixed these, and run a spelling + grammar checker on the rest of the paper
>
> > "While much of the amazing work" --- too subjective for an academic paper.
>
> We have revised this to read "much of the prior work."

---

> > ### Comment · Reviewer_MFVx · 2024-08-12
> > **Thank you for the response.**
> >
> > Thank you for the detailed response by the authors, which helped me to understand the paper better. With the new experimental results comparing against reward inference baselines, I am happy to increase my score to a 5. However, given the substantial improvements to the presentation that remain to be made, and the mismatch between the theoretical results and the practical algorithm, I am not currently comfortable raising my score beyond that.
> >
> > Some further comments on the presentation and theory-algorithm connection that I hope will help improve future versions of the paper:
> >
> > >  In the context of this paper, we look at a restricted set of skills: those defined by (open-loop) sequences of actions, $a^H_t$. This is why we use $a^H_t$ in our mutual information objective.
> >
> > Thank you for this explanation. This seems like a crucial piece of information that connects Lemma 2 to the algorithm actually used by the paper. I would strongly recommend emphasizing this point if you're going to keep the theoretical result in future versions of the paper, and replacing all instances of $z$, so as to make the connection clear. In addition, I would be careful to distinguish the mutual information $I(s^+, a^H_{1:t})$ between future states and open-loop *sequences* of actions and the mutual information $I(s^+, a^H_{t})$ with the human's action at a particular timestep $t$. It's not immediately obvious to me how these are related to each other (I assume the former is a summation of the latter?), and showing this connection is important for clarity if you're going to define a skill $z$ as a sequence $a^H_{1:t}$.
> >
> > >  In summary: yes, the theory differs from the practical algorithm, but the difference corresponds to a different choice of discounting function.
> >
> > Based on the author's response, it seems to me that the theory currently differs from the algorithm not just in the choice of discounting function, but in three crucial ways that should all be explicitly acknowledged and addressed in future revisions:
> > 1. The difference in the choice of discounting function (i.e. the difference between maximizing a discounted sum of per-step mutual information, as opposed to directly maximizing the mutual information).
> > 2. Whether the per-step mutual information is conditioned on the current state $s_t$ (in the form $I(s^+, a^H_t | s_t)$, which is the version used in Eq. 1) or not conditioned (in the form $I(s^+, a^H_t)$, which is the version considered in the theory).
> > 3. The fact that the ESR successor representation $\phi(s, a^H, a^R)$ conditions on $a^R$, not just $a^H$, leading it to maximize $I(s^+, a^H_t | s_t, a^R_t)$ instead of $I(s^+, a^H_t | s_t)$.
> >
> > The presence of these three differences make it hard to understand the applicability of the theoretical results to the algorithm. Ideally, the theory should be revised or generalized so that the theory is directly relevant to the algorithm. Minimally, some explanation and intuition should be provided for why we should expect the theory to generalize (even if this is not proven). Alternatively, as I suggested in my original review, it may be worth considering just focusing the paper on Section 4, and dropping the theoretical results in Section 3 altogether.
> >
> > > In our revision, we will add a derivation of the solution to our symmetrized infoNCE objective in Appendix D.
> >
> > In addition to adding this derivation to the Appendix, I would strongly recommend adding a Lemma or Proposition to Section 4 that states that the learned representations will converge to the desired mutual information metrics $I(s^+, a^H_t | s_t, a^R_t)$ and $I(s^+ | s_t, a^R_t)$. In other words, I would suggest restating Equations 8 and 9 as part of Lemma or Proposition, and then referring readers to the Appendix for a proof (along with a short explanation that this is what contrastive losses are designed to do).

---

> > > ### Author Response · Authors · 2024-08-12
> > > **Thank you for the suggestions**
> > >
> > > Thank you for the additional suggestions. We will make sure to incorporate these points in the final version.

---

> ### Author Response · Authors · 2024-08-07
> **References**
>
> Carroll, Micah, Rohin Shah, Mark K. Ho, Thomas L. Griffiths, Sanjit A. Seshia, Pieter Abbeel, and Anca Dragan. 2019. “On the Utility of Learning about Humans for Human-AI Coordination.” in _Conference on Neural Information Processing Systems_. arXiv.
>
> Eysenbach, Benjamin, Vivek Myers, Ruslan Salakhutdinov, and Sergey Levine. 2024. “Inference via Interpolation: Contrastive Representations Provably Enable Planning and Inference.”
>
> Hadfield-Menell, Dylan, Stuart J. Russell, Pieter Abbeel, and Anca Dragan. 2016. “Cooperative Inverse Reinforcement Learning.” _Advances in Neural Information Processing Systems_ 29.
>
> Poole, Ben, Sherjil Ozair, Aaron Van Den Oord, Alex Alemi, and George Tucker. 2019. “On Variational Bounds of Mutual Information.” in _International Conference on Machine Learning_. PMLR.
>
> Radford, Alec, Jong Wook Kim, Chris Hallacy, Aditya Ramesh, Gabriel Goh, Sandhini Agarwal, Girish Sastry, Amanda Askell, Pamela Mishkin, Jack Clark, Gretchen Krueger, and Ilya Sutskever. 2021. “Learning Transferable Visual Models From Natural Language Supervision.” in _International Conference on Machine Learning_. arXiv.

---

### Official Review · Reviewer_drkz · 2024-07-13

**Soundness:** 3
**Presentation:** 3
**Contribution:** 2
**Rating:** 5
**Confidence:** 3

**Summary:**

The authors propose a new training objective that motivates the agent to assist humans by maximizing their empowerment, agnostic of their rewards. The proposed empowerment objective is derived from mutual information between human actions and future states, which is estimated via contrastive representation learning and can be formulated as an RL reward. The authors empirically show that this training objective improves performances in a grid world game and the overcooked game over existing baselines.

**Strengths:**

- The writing of the paper is clear and easy to follow.
- The proposed method is novel to my knowledge.
- The authors have shown theoretical insights into their definition of empowerment to a certain degree.

**Weaknesses:**

- My main concern is the lack of baselines. The only baselines that the author chose to compare are AvE, which is another empowerment-based method that this work is based on, and random. From the results, AvE shows to be weaker even than random in most of the scenarios, which makes it less appealing as a baseline. At least for the Overcooked domain, there are plenty of baselines in [1] to choose from (or at least argue why they are not chosen).
- Following the previous point, even though there may not be enough empowerment-based methods to compare, the authors can still provide ablation studies to justify the design choices. As of now, the empirical result feels thin to me.
- The authors could provide more insights in the experiment section. For example, how well does the proposed method estimate empowerment? Or qualitatively,  what does the agent do to increase empowerment?  More analysis can help us better understand the method.

[1] Carroll, Micah et al. “On the Utility of Learning about Humans for Human-AI Coordination.”

**Questions:**

Figure 2 confuses me. I would appreciate a better explanation of what each of these images is referring to.

**Limitations:**

The authors have addressed the limitations and potential negative societal impact.

---

> ### Author Rebuttal · Authors · 2024-08-07
>
> Dear Reviewer,
>
> Thanks for the detailed review and suggestions for improving the paper. To address the main concern about baselines, we have added two additional baselines (see below). We have also run additional ablation experiments, and tried to address the other concerns in the discussion below. **Does this fully address the reviewer's concerns?**
>
> Kind regards,
>
> The authors.
>
>
> > My main concern is the lack of baselines.
>
> We have added a goal inference baseline based on Du et al. (2021) to the obstacle environment and a reward inference baseline using IQ-Learn (Garg et al., 2021) for both the obstacle environments and the overcooked settings. Our method outperforms both of these baselines across the environments studied. These experiments are included in the attached PDF.
>
> Regarding the baselines from Carrol et al. (2019) ([1] here): all of these require the true human reward. In contrast, our method does not assume access to the ground truth reward, and we aim to learn a collaborative policy through our empowerment objective.
>
> > provide ablation studies to justify the design choices.
>
> As suggested by the reviewer, we have run additional ablation experiments, studying the effective of the contrastive objective and the empowerment parametrization (see Fig. 1 in the rebuttal PDF).
>
> > qualitative analysis
>
> We have added additional qualitative results in the PDF.
> In the obstacle environment ESR learns to remove obstacles along the path the human is moving and in the Overcooked setting it learns collaborative behaviors like moving the plate to places the human can reach and placing onions in the pot. We will revise the paper with visualizations of some of these behaviors.
>
>
> > Figure 2 confuses me.
>
> We will clarify this figure caption in the revised paper. The goal of this figure is to relate the empowerment objective (Eq. 1) to the analysis of skill discovery (Lemma 2). On the left (a), we visualize how pairs of interacting policies induce a distribution over the state space, seen here as points on an $|\mathcal{S}|$-simplex. In the center (b), the orange polygon depicts the set of state distributions that the human's policy can attain when working with a fixed robot assistant. The black lines correspond to Eq 3 (Lemma 1), which say that empowerment relates to the ``diameter'' of this polygon.  To the right (c), we show how our empowerment objective corresponds to maximizing the size of this polytope, i.e., maximizing the human’s ability to control the distribution over state distributions in the environment. We will revise the figure to clarify.

---

> > ### Comment · Reviewer_drkz · 2024-08-13
> >
> > Thank you for the reply and the additional experiments. I am more inclined towards accepting now.

---

> ### Author Response · Authors · 2024-08-07
> **References**
>
> Carroll, Micah, Rohin Shah, Mark K. Ho, Thomas L. Griffiths, Sanjit A. Seshia, Pieter Abbeel, and Anca Dragan. 2019. “On the Utility of Learning about Humans for Human-AI Coordination.” in _Conference on Neural Information Processing Systems_. arXiv.
>
> Du, Yuqing, Stas Tiomkin, Emre Kiciman, Daniel Polani, Pieter Abbeel, and Anca Dragan. 2020. “AvE: Assistance via Empowerment.” Pp. 4560–71 in _Advances in Neural Information Processing Systems_. Vol. 33. Curran Associates, Inc.
>
> Garg, Divyansh, Shuvam Chakraborty, Chris Cundy, Jiaming Song, and Stefano Ermon. 2021. “IQ-Learn: Inverse Soft-Q Learning for Imitation.” Pp. 4028–39 in _Advances in Neural Information Processing Systems_. Vol. 34. Curran Associates, Inc.

---

### Official Review · Reviewer_PN73 · 2024-07-13

**Soundness:** 4
**Presentation:** 4
**Contribution:** 3
**Rating:** 7
**Confidence:** 4

**Summary:**

The paper addresses the problem setting of human-agent collaboration where the agent learns to empower human decision-making to exert greater control over the environment. By connecting empowerment to reward maximization, the paper proposes the ESR method, which learns an intrinsic reward function based on learned representations of future states. Experiments demonstrate that ESR outperforms the AvE baseline in a gridworld task and Overcooked.

**Strengths:**

1. To the best of my knowledge, the ESR method is novel. ESR provides the new insight of connecting empowerment and reward maximization. This allows using empowerment as an intrinsic reward for training agents with RL.

1. The paper shows ESR greatly outperforms the AvE baseline on more complex obstacle gridworlds and in Overcooked. In more complex environments, AvE is only able to achieve near-random performance. Results are also validated over multiple random seeds.

1. The code is provided to reproduce results.

**Weaknesses:**

1. The paper only shows results on 2 of the Overcooked environments. This decision is not justified and with 15-20 seeds on the main Overcooked results, it is likely not an issue of sufficient compute to run these experiments. Even if the performance is worse on the remaining environments, the paper should also report numbers on the full Overcooked setting with "Asymmetric Advantages", "Forced Coordination" and "Counter Circuit".

1. While the method addresses the problem of coordination without knowledge of the human's goal, the experiments demonstrating this are contrived. In Overcooked, the goal is fixed per task setting and is serving a dish. While ESR outperforms AvE, it greatly underperforms baselines from population-based training in [1]. This shows the large gap in empowerment versus the underlying objective of the task. The paper does not show results in a setting where the goals and preferences of the human are complex to communicate.

1. The description of AvE lacks suffcient detail given it is the only baseline and addresses the same problem. Why does AvE struggle in more complex environments? How does it differ from ESR?

Minor:

1. The acronym ESR is not defined in the main text and is only defined in the caption of Algorithm 1.

References:

1. Carroll, Micah, et al. "On the utility of learning about humans for human-ai coordination." Advances in neural information processing systems 32 (2019).

**Questions:**

1. L160 refers to "the CIRL setting", but CIRL is never defined. What does CIRL stand for?

1. As stated in the limitations, ESR assumes access to the human actions to learn the empowerment reward. Can ESR work with noisy predictions of human actions?

1. What is the performance of methods that assume access to the human intent via the task reward function in Overcooked? Are the results in Figure 5 directly comparable to those in Figure 4 from the Overcooked paper?

1. Is the assumption that the agents share the same state space central to the empowerment learning objective? Is it possible to implement ESR if both agents operate from separate egocentric views?

**Limitations:**

Yes, the paper discusses the limitations and safety risks in Section 6.

---

> ### Author Rebuttal · Authors · 2024-08-07
>
> Dear Reviewer,
>
> Thanks for the review, and the suggestions for improving the paper. As suggested by the reviewer, we have evaluated the proposed method on a number of additional tasks, and compared with a number of additional baselines. **Together with the responses below, does this fully address the reviewer's concerns? About the paper?**
>
> Kind regards,
>
> The authors.
>
>
> > Evaluating on additional Overcooked environments
>
>
> To expand our experimental results, we have run an additional experiment on the “asymmetric advantage” layout in Overcooked (see Table 1 in the attached pdf). We initially excluded this setting due to a lack of good models for human collaboration to evaluate on: policies that imitate expert data and/or use deep RL with self-play struggle to robustly learn collaborative behavior without additional structure. Our new result shows ESR can outperform baselines when playing with a heuristic-planning human model based on Carroll et al. (2019).
>
> > While the method addresses the problem of coordination without knowledge of the human's goal, the experiments demonstrating this are contrived. In Overcooked, the goal is fixed per task setting and is serving a dish.
>
> We will add discussion of more complex settings that would be enabled by our method to the future work section. Examples of realistic settings where empowerment objectives could be effective include copilot-style assistants (Dohmke, 2022) and biomedical devices that improve human agency (Bryan et al., 2023; Pan et al., 2024). We note that the Overcooked environment has been used as a primary evaluation for human-AI collaboration in numerous works (Carroll et al., 2019; Hong et al., 2023; Knott et al., 2021; Laidlaw & Dragan, 2022; Lauffer et al., 2023; Strouse et al., 2021).
>
>
> > The paper does not show results in a setting where the goals and preferences of the human are complex to communicate.
>
>
> This is a good point, the true reward of the Overcooked setting is easily communicated. However, our focus is on learning a collaborative policy without explicitly communicating the true reward, or the method of collaboration, which is more difficult to express.
>
>
> > The description of AvE lacks sufficient detail given it is the only baseline and addresses the same problem. Why does AvE struggle in more complex environments? How does it differ from ESR?
>
>
> The key limitation of AvE is it tries to approximate the empowerment quantity which they define as $\max_\pi I(s^+;a_t\mid s_t)$ through random rollouts. This is completely intractable in high-dimensional environments (as our larger experiments show), where random rollouts don’t produce meaningful behaviors.
>
> Additionally, since the rollouts select actions uniformly at random, even in the limit of infinite computation, the AvE objective doesn’t actually compute empowerment with respect to meaningful human behaviors (either under their definition or our definition). Intuitively, imagine a human performing some task in a room with thin walls. The AvE objective would incentivize the robot knocking down the walls to maximize the human’s ability to leave the house, even though they have no desire to do so. Our ESR objective would help them with the task at hand.
>
>
>
> > The acronym ESR is not defined in the main text and is only defined in the caption of Algorithm 1.
>
> We have fixed this.
>
>
> > L160 refers to "the CIRL setting", but CIRL is never defined. What does CIRL stand for?
>
> CIRL refers to the cooperative inverse reinforcement learning setup from Hadfield-Menell et al. (2016), in which an assistant cooperates with a human without knowing the true human reward.
>
>
> > As stated in the limitations, ESR assumes access to the human actions to learn the empowerment reward. Can ESR work with noisy predictions of human actions?
>
> In general, the ESR objective under an arbitrary (state-independent) noise model becomes a lower bound on the “true” ESR empowerment (applying the data processing inequality to the Markov chain $\hat{a}\to a\to s^+$). This suggests that ESR will be conservative in the presence of noisy action observations.
>
>
> > While ESR outperforms AvE, it greatly underperforms baselines from population-based training in [1].
>
> Methods that assume access to the true reward (i.e., Figure 4 of the Overcooked paper (Carroll et al., 2019)) will in general perform better than methods such as ESR that do not.
>
> > Is the assumption that the agents share the same state space central to the empowerment learning objective? Is it possible to implement ESR if both agents operate from separate egocentric views?
>
> While we do make this assumption in our formulation, in many cases it may be possible to use the same algorithm in partially observed settings. If the human has partial observability, this corresponds to maximizing empowerment of a suboptimal human (it throws away information about the state in its observation), which will often nevertheless be desirable. If the agent has partial observability, the empowerment objective will only increase the human’s influence over the parts of the environment the agent can observe. In many cases this may be fine, but in environments where the agent can restrict the information in its own observations, care must be taken to mitigate reward hacking (Amodei et al., 2016)—in this case, situations where the agent can choose to look at only the areas the human has influence over. We will add discussion on these points to the limitations and future work section.

---

> ### Author Response · Authors · 2024-08-07
> **References**
>
> Amodei, Dario, Chris Olah, Jacob Steinhardt, Paul Christiano, John Schulman, and Dan Mané. 2016. “Concrete Problems in AI Safety.”
>
> Bryan, Matthew J., Linxing Preston Jiang, and Rajesh P N Rao. 2023. “Neural Co-Processors for Restoring Brain Function: Results from a Cortical Model of Grasping.” _Journal of Neural Engineering_ 20(3).
>
> Carroll, Micah, Rohin Shah, Mark K. Ho, Thomas L. Griffiths, Sanjit A. Seshia, Pieter Abbeel, and Anca Dragan. 2019. “On the Utility of Learning about Humans for Human-AI Coordination.” in _Conference on Neural Information Processing Systems_. arXiv.
>
> Dohmke, Thomas. 2022. “GitHub Copilot Is Generally Available to All Developers.” _Retrieved July_ 25:2023.
>
> Du, Yuqing, Stas Tiomkin, Emre Kiciman, Daniel Polani, Pieter Abbeel, and Anca Dragan. 2020. “AvE: Assistance via Empowerment.” Pp. 4560–71 in _Advances in Neural Information Processing Systems_. Vol. 33. Curran Associates, Inc.
>
> Hadfield-Menell, Dylan, Stuart J. Russell, Pieter Abbeel, and Anca Dragan. 2016. “Cooperative Inverse Reinforcement Learning.” _Advances in Neural Information Processing Systems_ 29.
>
> Hong, Joey, Sergey Levine, and Anca Dragan. 2023. “Learning to Influence Human Behavior with Offline Reinforcement Learning.” in _Conference on Neural Information Processing Systems_. arXiv.
>
> Knott, Paul, Micah Carroll, Sam Devlin, Kamil Ciosek, Katja Hofmann, A. D. Dragan, and Rohin Shah. 2021. “Evaluating the Robustness of Collaborative Agents.” in _AAMAS_. arXiv.
>
> Laidlaw, Cassidy, and Anca Dragan. 2022. “The Boltzmann Policy Distribution: Accounting for Systematic Suboptimality in Human Models.” in _International Conference on Learning Representations_. arXiv.
>
> Lauffer, Niklas, Ameesh Shah, Micah Carroll, Michael D. Dennis, and Stuart Russell. 2023. “Who Needs to Know? Minimal Knowledge for Optimal Coordination.” Pp. 18599–613 in _Proceedings of the 40th International Conference on Machine Learning_. PMLR.
>
> Pan, Michelle, Mariah Schrum, Vivek Myers, Erdem Bıyık, and Anca Dragan. 2024. “Coprocessor Actor Critic: A Model-Based Reinforcement Learning Approach For Adaptive Brain Stimulation.” in _International Conference on Machine Learning_.
>
> Strouse, Dj, Kevin McKee, Matt Botvinick, Edward Hughes, and Richard Everett. 2021. “Collaborating with Humans without Human Data.” Pp. 14502–15 in _Advances in Neural Information Processing Systems_. Vol. 34. Curran Associates, Inc.

---

### Official Review · Reviewer_FYis · 2024-07-15

**Soundness:** 4
**Presentation:** 3
**Contribution:** 2
**Rating:** 6
**Confidence:** 4

**Summary:**

This paper studies the problem of human-AI collaboration. The goal is to train an cooperative policy that can work with human together in the environment to achieve a shared goal.

The key idea of this paper is to maximize the influence of the human's actions on the environment, which is called empowerment. The paper provides an information geometric interpretation of empowerment and develops a algorithm based on SAC for estimating and optimizing empowerment. This method does not require to infer human's reward model.

**Strengths:**

1. Code is provided.
2. The method removes the need to infer human reward function, making it scalable to high-dim space.
3. The method is theoretically grounded and is simple to implement.

**Weaknesses:**

* No experiments involving real human subjects are conducted.
* As suggested by the limitation, it's infeasible to train such a policy with a real human subject, especially in the task like overcooked which takes 1.5M steps to train...
* The paper only compares to one baseline AvE. As the overcooked is a widely used environment studying human-robot collaboration, more results are welcome.

**Questions:**

* What's the motivation by sampling k from a Geometric distribution in Line 136?
* IIUC, the learned policy will "overfit" to the human policy which it is trained with? How to scale up the system to utilize the (offline) data from different human policy with different skills / characteristics / optimality?
* Is there any measurement available to measure the "human-compatibility" during training? That is, will the human subject feel better with some robot policies against other robot policies even though the learning converge in the same speed?
* Some clarity issues.
  * Typo Line 268 "limit’s".
  * No closing parenthesis in Line 218.
  * What is "this set" in Line 142 (if the reader don't read caption of Fig2)

**Limitations:**

The limitations are well addressed.

---

> ### Author Rebuttal · Authors · 2024-08-07
>
> Dear Reviewer,
>
> We thank the reviewer for the comments and suggestions for improving the paper. It seems like the reviewer's main suggestions were to add additional baselines, which we have done by adding comparisons with a goal inference method and a reward inference method. **Together with the discussion below, does this fully address the reviewer's concerns?**
>
> Kind regards,
>
> The authors.
>
> > The paper only compares to one baseline AvE. As Overcooked is a widely used environment studying human-robot collaboration, more results are welcome.
>
> We have added a goal inference baseline (Du et al., 2021) to the obstacle environment and a reward inference baseline using IQ-Learn (Garg et al., 2021) for both the obstacle environments and the overcooked settings. Our method outperforms both of these baselines across the environments studied. We also add additional ablations of our approach (contrastive loss and parameterization). The two ablations are with the norm and diff variants of our reward. The norm reward computes the reward as the difference of norms between phi(s,a) and phi(s). The diff reward computes the reward as the norm of the difference between phi(s, a) and phi(s). These experiments are included in the attached PDF. Note that a key difference between our evaluations and prior work that studies assistance in Overcooked (e.g., Carroll et al. (2019)) is that we do not assume any access to the human reward *or* a model of the environment / possible reward structures.
>
>
> > Is there any measurement available to measure the "human-compatibility" during training?
>
> Thanks for this suggestion! This paper is primarily mathematical and algorithmic and nature, and so we'll provide an mathematical/algorithmic answer: Eq. 6 says that we can measure the degree of (minimax) compatibility via the mutual information. We have run an additional experiment to plot this mutual information throughout training (see Fig. 4 in rebuttal PDF). Visualizing the learned agent, we see that the agent does indeed become more helpful as this mutual information increases.
>
> Of course, the gold standard in human-AI interaction is human user studies, which are beyond the scope of this mathematical/algorithmic paper.
>
> > IIUC, the learned policy will "overfit" to the human policy which it is trained with? How to scale up the system to utilize the (offline) data from different human policy with different skills / characteristics / optimality?
>
> As with any machine learning algorithm, there is a risk of overfitting when learning from limited data, and learning from a larger quantity of data may mitigate this risk. While our study focuses on the online learning of collaborative policies, the contrastive empowerment objective could also be applied to an offline dataset. In the case where diverse offline data is limited, it may be beneficial to initialize this contrastive policy with a large pretrained model that can impose strong inductive biases on the successor representations. Future work could also explore how multi-modal models (e.g., Chen et al., 2020) combined with our method to further boost performance
>
> > What's the motivation by sampling k from a Geometric distribution in Line 136?
>
> The choice of distribution for $k$ dictates the ``horizon'' of the human's empowerment: do we want the human's actions to have a high influence over the outcomes in the next hour, or over the outcomes in the next day. For simplicity, we used a Geometric distribution to be consistent with RL standards (where we care about rewards accumulated under a geometric distribution) and set the parameter to the same value as used by the underlying RL algorithm.
>
> > No experiments involving real human subjects are conducted.
>
> We have revised the paper to mention this limitation in the conclusion. Many prior papers also aim to establish the algorithmic foundations for human-AI algorithms before proceeding with human studies (Chan et al., 2019; He et al., 2023; Ngo et al., 2024; Pan et al., 2022; Ratner et al., 2018; Robertson et al., 2023; Zhuang & Hadfield-Menell, 2020).
>
> > it's infeasible to train such a policy with a real human subject, especially in the task like overcooked which takes 1.5M steps to train…
>
> Yes, our method needs a large quantity of human data. However, we could apply our method in the offline setting where we have large scale existing datasets for this purpose (see e.g., Xie et al. (2018)). We could then fit the ESR features to the dataset and use the empowerment reward with an offline RL algorithm.
>
> > Typo Line 268 "limit’s".
>
> We have fixed this.
>
> > No closing parenthesis in Line 218.
>
> We have fixed this.
>
> > What is "this set" in Line 142 (if the reader don't read caption of Fig, 2)
>
> We have clarified this.

---

> > ### Comment · Reviewer_FYis · 2024-08-09
> >
> > Thanks for the response.
> >
> >
> > 1. I can't see Fig. 4 in rebuttal PDF.
> >
> > 2. "While our study focuses on the online learning of collaborative policies, the contrastive empowerment objective could also be applied to an offline dataset." The purpose for this question is how you address the problem when using offline dataset that it might contains a lots of different human subjects. Here I am not questing about the "overfitting" but the potential issue of "overfitting to a single human subject". If you want to use offline dataset, should we always ask a specificed human subject to collect data?

---

> ### Author Response · Authors · 2024-08-07
> **References**
>
> Chan, L., Hadfield-Menell, D., Srinivasa, S., & Dragan, A. (2019). The assistive multi-armed bandit. _ACM/IEEE International Conference on Human-Robot Interaction_. HRI.
>
> Chen, L., Paleja, R., Ghuy, M., & Gombolay, M. (2020). Joint goal and strategy inference across heterogeneous demonstrators via reward network distillation. In Proceedings of the 2020 ACM/IEEE international conference on human-robot interaction (pp. 659-668).
>
> Du, Y., Tiomkin, S., Kiciman, E., Polani, D., Abbeel, P., & Dragan, A. (2020). AvE: Assistance via Empowerment. _Advances in Neural Information Processing Systems_, _33_, 4560–4571.
>
> Garg, D., Chakraborty, S., Cundy, C., Song, J., & Ermon, S. (2021). IQ-Learn: Inverse soft-Q Learning for Imitation. _Advances in Neural Information Processing Systems_, _34_, 4028–4039.
>
> Hadfield-Menell, D., Russell, S. J., Abbeel, P., & Dragan, A. (2016). Cooperative inverse reinforcement learning. _Advances in Neural Information Processing Systems_, _29_.
>
> He, J. Z.-Y., Brown, D. S., Erickson, Z., & Dragan, A. (2023). Quantifying assistive robustness via the natural-adversarial frontier. _Proceedings of The 7th Conference on Robot Learning_, 1865–1886.
>
> Ngo, R., Chan, L., & Mindermann, S. (2024). The alignment problem from a deep learning perspective. _The Twelfth International Conference on Learning Representations_.
>
> Pan, A., Bhatia, K., & Steinhardt, J. (2022). The effects of reward misspecification: Mapping and mitigating misaligned models. _International Conference on Learning Representations_. ICLR.
>
> Ratner, E., Hadfield-Menell, D., & Dragan, A. D. (2018). Simplifying reward design through divide-and-conquer. _Robotics - Science and Systems_. Robotics - Science and Systems.
>
> Robertson, Z., Zhang, H., & Koyejo, S. (2023). Cooperative inverse decision theory for uncertain preferences. _Proceedings of The 26th International Conference on Artificial Intelligence and Statistics_, 5854–5868.
>
> Xie, D., Shu, T., Todorovic, S., & Zhu, S.-C. (2018). Learning and Inferring “Dark Matter” and Predicting Human Intents and Trajectories in Videos. _IEEE Transactions on Pattern Analysis and Machine Intelligence_, _40_(7), 1639–1652.
>
> Zhuang, S., & Hadfield-Menell, D. (2020). Consequences of misaligned AI. _Advances in Neural Information Processing Systems_, _33_, 15763–15773.

---

> ### Author Response · Authors · 2024-08-09
>
> 1. Apologies for the oversight, we have added the mutual information and accuracy as a table in the official comment above, and will include a line plot of it in the final paper. As the empowerment policy improves, the human's actions take on more influence over the future states, increasing the mutual information.
>
> 2. Thank you for the clarification—this is a great question. Learning to empower the human, in contrast to learning the human's reward, is less sensitive to the choice of human policy, and can even benefit from a wide variety of human data to train on. The ESR policy learns to take actions that make the human's choice of action have a high impact on the future state. If the human's policy changes—i.e., it takes a different action in the same state—the empowering policy will still empower the human to reach their goal, irrespective of the human subject. On our new "asymmetric advantage" environment results, we trained the ESR agent against an ensemble of three goal-directed human "personas", each with their own, distinct behavior and method of collaboration. We found that this diversity was important to good performance, and we hypothesize that this is because the assistant must learn to pay attention to the human's action in order to predict the future state well.
> While these experiments were performed online, we reason that the same principles would apply to offline data. Data from diverse policies should help empowerment.
>
> **Do these responses address your concerns?**

---

> > ### Comment · Reviewer_FYis · 2024-08-12
> >
> > Thank you for your follow-up comment. It greatly mitigates my concerns and now I am more inclined to accept the paper.

---

### Author Rebuttal · Authors · 2024-08-07

We would like to thank the reviewers for their feedback. Reviewers mentioned concerns about baselines and presentation, which we have responded to in detail below. Based on this feedback, we have run additional ablations and baselines for our method and conducted additional qualitative analysis (see attached PDF).

---

> ### Author Response · Authors · 2024-08-08
> **Evaluation on Additional Overcooked Environment**
>
> We have added Table 1 to our revision, comparing (reward-unsupervised) assistants collaborating with an expert human model on an additional Overcooked layout. Our method outperforms the new "reward inference" baseline as well as "AvE" and "random" on all layouts tested. Results are shown with one standard error.
>
> #### Table 1: Additional evaluation on the “asymmetric advantages” Overcooked layout
> |Layout| **ESR (Ours)** | Reward Infer | AvE | Random |
> |---|---|---|---|---|
> | Asymmetric Advantages |  $\mathbf{72.00 \pm 5.37}$ | $60.33 \pm 0.26$ | $36.71 \pm 1.71$ | $59.36 \pm 1.07$ |
> | Coordination Ring | $\mathbf{8.40 \pm 0.69}$ | $5.96 \pm 0.20$ | $5.69 \pm 0.93$ | $6.02$ |
> | Cramped Room | $\mathbf{91.33 \pm 4.08}$ | $39.24 \pm 0.35$ | $5.13 \pm 1.31$ | $69.26$ |

---

> ### Author Response · Authors · 2024-08-09
> **Visualizing Mutual Information Throughout Training**
>
> We have added Figure 4 to our revision, summarized here as a table, showing the mutual information and accuracy of the model throughout training on the "asymmetric advantages" Overcooked layout. The accuracy measures the model's ability to distinguish a true future state from other, random future states given the current state and action. Mutual information is measured between the human's action and the future state, signifying that as training progresses, the human's action takes on more influence over the future states.
>
> |**Step**| 0K | 200k | 400k | 600k | 800k | 1M |
> |---|---|---|---|---|---|---|
> |**Mutual Information**| 0.0 | 0.72 | 0.37 | 0.54 | 0.60 | 0.79 |
> |**Accuracy (%)**| 0.0 | 51.4 | 62.4 | 67.2 | 69.4 | 70.7 |

---

### Decision · Program_Chairs · 2024-09-25

**Decision:**

Accept (poster)

**Comment:**

This paper initially received a two borderline accepts (FYis, drkz), an accept (PN73), a borderline reject (MFVx), and a reject (KT7u). All reviewers appreciated the novelty of building assistive agents that improve human empowerment, particularly in higher dimensional spaces, and the theoretical contributions supporting this. Beyond requesting clarifications, the reviewers raised concerns about baselines and the sufficiency of empirical results (including lack of human evaluation).

In the discussion phase, the authors addressed most of these concerns by adding new baselines to their benchmarks, presenting new results on the asymmetric advantage layout of overcooked, including additional ablations for design choices, and clarifying the assumptions and scope of their work. Based on this, reviewers FYis, MFVx, and KT7u increased overall ratings, and drkz expressed they are more inclined to accept the paper.

The AC concurs with the reviewers on the novelty of this work and recommends acceptance. While human study & empirical results of human compatibility would strengthen this work, the paper’s current theoretical and empirical contributions hold significant value. The authors are encouraged to improve the paper’s readability for the final version with a systematic introduction of new terminology (PN73), edits to figures (drkz), and rigorous mathematical formalization (MFVx) -- based on the constructive discussion between the authors and the reviewers.